# A Sec14-like phosphatidylinositol transfer protein paralog defines a novel class of heme-binding proteins

Danish Khan[1], Dongju Lee[2†], Gulcin Gulten[1†], Anup Aggarwal[1], Joshua Wofford[3,4], Inna Krieger[1], Ashutosh Tripathi[2], John W Patrick[3], Debra M Eckert[5], Arthur Laganowsky[3], James Sacchettini[1], Paul Lindahl[1,3], Vytas A Bankaitis[1,2,3]*

[1]Department of Biochemistry and Biophysics, Texas A&M University, College Station, United States; [2]Department of Molecular and Cellular Medicine, Texas A&M Health Sciences Center, College Station, United States; [3]Department of Chemistry, Texas A&M University, College Station, United States; [4]Department of Chemistry, Charleston Southern University, North Charleston, United States; [5]Department of Biochemistry, University of Utah School of Medicine, Salt Lake City, United States

*For correspondence:
vytas@tamu.edu

†These authors contributed equally to this work

Competing interests: The authors declare that no competing interests exist.

**Abstract** Yeast Sfh5 is an unusual member of the Sec14-like phosphatidylinositol transfer protein (PITP) family. Whereas PITPs are defined by their abilities to transfer phosphatidylinositol between membranes in vitro, and to stimulate phosphoinositide signaling in vivo, Sfh5 does not exhibit these activities. Rather, Sfh5 is a redox-active penta-coordinate high spin $Fe^{III}$ hemoprotein with an unusual heme-binding arrangement that involves a co-axial tyrosine/histidine coordination strategy and a complex electronic structure connecting the open shell iron $d$-orbitals with three aromatic ring systems. That Sfh5 is not a PITP is supported by demonstrations that heme is not a readily exchangeable ligand, and that phosphatidylinositol-exchange activity is resuscitated in heme binding-deficient Sfh5 mutants. The collective data identify Sfh5 as the prototype of a new class of fungal hemoproteins, and emphasize the versatility of the Sec14-fold as scaffold for translating the binding of chemically distinct ligands to the control of diverse sets of cellular activities.

## Introduction

Lipid metabolism is a major component of the eukaryotic strategy for organizing and regulating cell signaling. This involvement is exemplified by the metabolism of phosphatidylinositol (PtdIns) and its phosphorylated derivatives – the phosphoinositides. Even though eukaryotes only produce five to seven chemically distinct phosphoinositide classes (*Balla, 2013*; *De Craene et al., 2017*; *Wallroth and Haucke, 2018*; *Dickson and Hille, 2019*), these lipids contribute to the control of a broad set of cellular processes. How such diversity of biological outcome is achieved from such a limited set of chemical codes is not well understood. New insights into the mechanisms by which biological outcomes of phosphoinositide signaling are diversified come from studies of how PtdIns-4-phosphate (PtdIns4P) signaling is channeled to specific biological outcomes. In that case, the PtdIns 4-OH kinases are biologically insufficient enzymes that are unable to generate sufficient PtdIns4P to override the action of erasers of PtdIns4P signaling, such as lipid phosphatases and PtdIns4P-sequestering activities, to support productive signaling. These deficiencies are overcome by the action of Sec14-like PtdIns transfer proteins (PITPs) that stimulate PtdIns 4-OH kinase activities so that sufficient PtdIns4P pools are generated in a precisely regulated manner (*Ile et al., 2006*). It is through this general strategy that individual Sec14-like PITPs channel PtdIns 4-OH kinase activities to specific

biological outcomes (*Schaaf et al., 2008*; *Bankaitis et al., 2010*; *Nile et al., 2010*; *Grabon et al., 2019*).

The biological importance of PITPs in regulating phosphoinositide metabolism is demonstrated by the expansion of the Sec14-like PITPs into the large Sec14 protein superfamily united by a conserved Sec14-fold (*Sha et al., 1998*; *Saito et al., 2007*; *Schaaf et al., 2008*; *Huang et al., 2016*), and by the striking phenotypes associated with loss-of-function of individual PITPs in biological systems that range from yeast (*Bankaitis et al., 1989*; *Wu et al., 2000*; *Ren et al., 2014*; *Huang et al., 2018*), to plants (*Peterman et al., 2004*; *Vincent et al., 2005*; *Ghosh et al., 2015*; *Huang et al., 2016*), and to vertebrates (*Hamilton et al., 1997*; *Alb et al., 2003*; *Kono et al., 2013*). Sec14-like PITPs potentiate activities of PtdIns 4-OH kinases via a heterotypic lipid exchange cycle between a membrane-docked PITP molecules and the bilayer where the dynamics of an abortive exchange of a 'priming' lipid for PtdIns exposes the PtdIns to the lipid kinase catalytic site. In this way, the lipid exchange cycle renders PtdIns a superior substrate for the enzyme. Structure-based predictions that PtdIns-binding is conserved across the Sec14 superfamily, while the nature of the priming lipid is diversified, suggests a mechanism where the PITP acts as a metabolic sensor for activating PtdIns4P signaling in response to a metabolic input (*Schaaf et al., 2008*; *Bankaitis et al., 2010*; *Grabon et al., 2019*).

As all characterized Sec14-like PITPs are lipid-binding proteins, interrogation of the priming ligand question has focused on lipids as it is widely presumed that lipid exchange/transfer activity is the common feature amongst these proteins. Recent work suggests the ligand menu for Sec14-like proteins might be broader than previously thought, however. Of the six Sec14-like PITPs expressed in yeast, Sfh5 exhibits several distinguishing properties. First, it is the most divergent member of the group as Sfh5 shares the lowest amino acid sequence homology (17.4% identity/33% similarity) with Sec14. Second, Sfh5 is the only yeast Sec14-like protein that exhibits low levels of PtdIns-transfer activity in vitro, and even high levels of Sfh5 expression fail to rescue the growth phenotypes associated with reduced activity of Sec14 in vivo (*Li et al., 2000*). Third, the electrostatic properties of Sfh5 are unique among the cohort of yeast Sec14-like PITPs as both the protein surface and large regions of the putative lipid-binding cavity exhibit significant electropositive character – suggesting Sfh5 does not bind a lipid (*Tripathi et al., 2019*).

Herein, we describe a biophysical and structural characterization of Sfh5. We report Sfh5 is a novel penta-coordinate high spin $Fe^{3+}$ heme-binding protein conserved across the fungal kingdom that exhibits unusual heme-binding properties. While the biological functions of Sfh5 remain to be determined, chemogenomic analyses forecast Sfh5 plays a role in organic oxidant-induced stress responses. As such, Sfh5 is the founding member of a new and conserved class of fungal hemoproteins, and provides the first demonstration that the spectrum of ligands for PITP-like proteins extends beyond lipids.

## Results

### The unconventional Sec14-like PITP Sfh5 is a heme-binding protein

The unusual biochemical properties ascribed to Sfh5 became particularly evident when the recombinant protein was overexpressed in bacteria. Within a few hours after induction, the producing bacteria acquired a distinct reddish color. Moreover, the purified recombinant $His_6$-Sfh5 itself exhibited an intense reddish-brown color which was resistant to dialysis or gel filtration – suggesting a tightly-bound transition metal (*Figure 1A*). Inductively coupled plasma mass spectrometry (ICP-MS) analysis of purified $His_6$-Sfh5 identified the bound metal as Fe, and quantification indicated an Fe/Sfh5 molar ratio of 0.33. Moreover, Sfh5 exhibited a Soret peak with an absorption maximum at 404 nm – the spectral signature of an $Fe^{3+}$ heme (*Figure 1B*). In support of this interpretation, sodium dithionite treatment of purified Sfh5 induced a red-shift of the Soret peak to 427 nm – signifying reduction to the $Fe^{2+}$ state. Reduction of purified recombinant Sfh5 in the context of pyridine hemochromagen assays revealed the formation of additional spectral peaks at 525 nm and 557 nm (*Figure 1B*, inset). Those peaks were diagnostic of β- and α-bands, respectively, of non-covalently bound heme *b* (*Ghosh et al., 2005*; *Beltrán et al., 2015*; *Barr and Guo, 2015*). From the spectral intensity measurements in those assays, ~30% of the purified Sfh5 was estimated to be loaded with heme *b* – a value consistent with the stoichiometry derived from ICP-MS analyses.

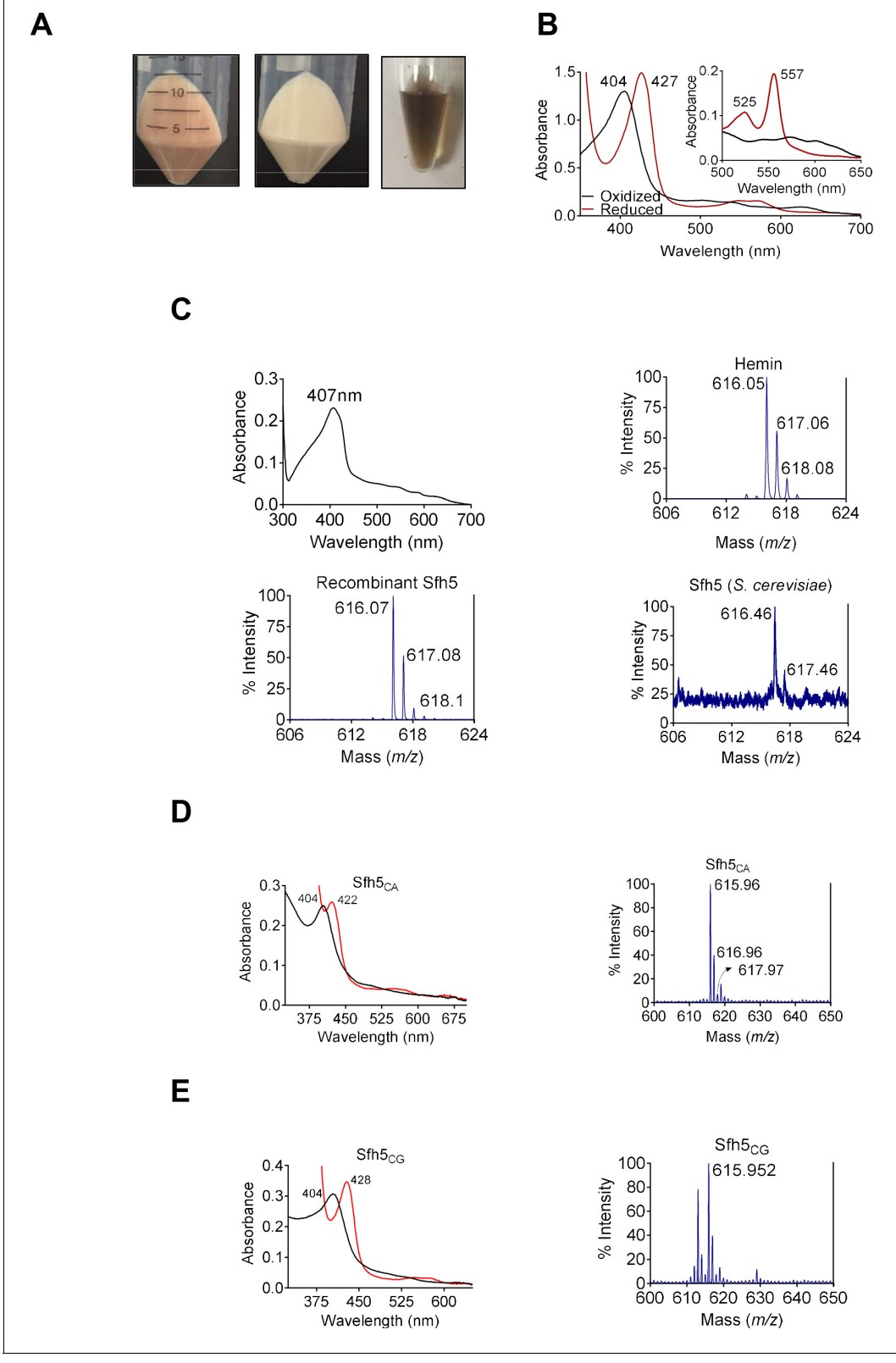

**Figure 1.** Sfh5 is a heme-binding protein. (**A**) Cell pellets of *E. coli* BL21(DE3) culture expressing Sfh5 are reddish brown color (left panel) relative to cell pellets of BL21(DE3) expressing Sec14 (middle panel). Panel at right shows an image of a purified recombinant Sfh5 (500 µM) solution which is distinguished by a deep reddish brown color. (**B**) UV-vis absorption spectrum of purified Sfh5 is traced in black. Spectra in red identify the spectral profile of Sfh5 after reduction with dithionite. Inset shows the 500 to 650 nm region of the UV-vis spectrum from pyridine hemochromagen assays. The

*Figure 1 continued on next page*

Figure 1 continued

indicated β and α bands at 525 and 557 nm, respectively, diagnose a noncovalently-bound heme *b*. (C) Top left: UV-vis absorption spectrum of a clarified cell lysate prepared from *E. coli* expressing Sfh5 absent an N' terminal 8xHis tag. The spectra were calibrated against a cell lysate prepared in parallel from mock-expressing *E. coli* cells (vector-only). Top left: The UV-vis absorption spectrum of clarified cell lysate prepared from *E. coli* BL21 (DE3) cells expressing tag-less Sfh5. Mock lysate prepared from *E. coli* BL21 (DE3) cells was used as blank to calibrate the spectrophotometer. Both *E. coli* cultures were induced with 100 μM IPTG and incubated overnight at 16 °C prior to harvesting and lysing cells. Top right: MALDI-TOF mass spectrum signature of hemin (positive standard), Bottom left and bottom right panels show the heme signatures of purified recombinant Sfh5 and Sfh5 purified directly from *S. cerevisiae*, respectively. Peaks at 616 ± 2 Da constitute a signature for protoporphyrin IX. Total spectral counts within the indicated m/z ranges were as follows: hemin, 30821; recombinant Sfh5, 8038; *S. cerevisiae* Sfh5, 672. (D, E) UV-vis absorption (left panels) and MALDI-TOF mass spectra (right panels) of purified recombinant Sfh5 sourced from *C. albicans* (Sfh5$^{CA}$) and *C. glabrata* (Sfh5$^{CG}$). Spectral profiles of dithionite-treated versions of these proteins were red-shifted to 422 and 428 nm for Sfh5$^{CA}$ and Sfh5$^{CG}$, respectively. Total counts within the indicated m/z ranges were 1404 and 1739 for Sfh5$^{CA}$ and Sfh5$^{CG}$, respectively.

The online version of this article includes the following figure supplement(s) for figure 1:

**Figure supplement 1.** Sfh5 oligomerization and the unit cell.

Several lines of evidence identify Sfh5 as a bona fide heme-binding protein. First, a tag-less version of Sfh5 expressed in *E. coli* showed similar Soret profiles – demonstrating that heme-binding by Sfh5 is not a His-tag artifact (*Figure 1C*; top left panel). Second, matrix-assisted laser desorption-ionization (MALDI) mass spectroscopy of recombinant His$_6$-Sfh5 purified from bacteria, as well as of His-tagged Sfh5 purified directly from yeast, yielded iron-protoporphyrin IX signatures at an $m/z = 616 \pm 2$ Da (*Figure 1C*; bottom panels). Third, the heme-binding property was conserved in Sfh5 orthologs from other fungal species – including pathogenic fungi from the genus *Candida*. Purified recombinant Sfh5 orthologs of *Candida albicans* (Sfh5$^{CA}$, orf 19.4897) and *Candida glabrata* (Sfh5$^{CG}$, CAGL0I05940g) were reddish-brown, both *Candida* Sfh5 preparations exhibited obvious Soret profiles, and both preparations yielded iron-protoporphyrin IX signatures when interrogated by MALDI mass spectrometry (*Figure 1D,E*). Since all but one of the *Candida* Sfh5 proteins lack Cys residues, the Sfh5 heme is not coordinated by cysteine – thereby excluding P450-type heme centers or [Fe-S] cluster proteins.

## Sfh5 crystal structure

Hydrodynamic analyses indicated that purified recombinant Sfh5 exists in a uniform oligomeric state calculated to be a dimer in solution (*Figure 1—figure supplement 1*), and crystallization conditions were optimized so that brown, rod-shaped, diffraction-quality crystals appeared within three days of incubation (*Figure 2A*). The crystallographic symmetry was of the orthorhombic space group P4$_3$2$_1$2 (unit cell dimensions a = b = 205 Å, c = 68 Å) with a high solvent content of ~64% (*Figure 2—figure supplement 1*). As molecular replacement proved unsuitable for solving the Sfh5 structure, single-wavelength anomalous diffraction was used to exploit the iron anomalous signal at Cu Kα edge to obtain phase information. The final Sfh5 structure was refined to 2.9 Å resolution (*Table 1*; PDB ID 6W32). The crystal asymmetric unit (ASU) contained three molecules of Sfh5 with a single heme-Fe bound to each of the chains. The crystallized form appears to be monomeric. While two molecules in the ASU make relatively close contact, the third molecule lacks a contact partner when checked across the symmetry axis (*Figure 2B*). Assessment of the crystal contact interfaces by the PISA server (https://www.ebi.ac.uk/pdbe/pisa/) concluded that these crystal contacts are unlikely to support stable quaternary assemblies in solution. One chain in the asymmetric unit (chain B) exhibited well-ordered electron density and was built with high confidence and good stereo-chemistry. However, the electron densities of the N-terminal domain of the two other chains (chains A and C) were poorly defined. Thus, the model for chain B was used for all subsequent structural analyses.

Sfh5 adopted a typical Sec14-fold comprised of eight α-helices, five β-strands, five short 3$_{10}$ helices and a β-hairpin turn (*Sha et al., 1998*; *Schaaf et al., 2008*; *Ren et al., 2014*). A tripod motif formed by helices A$_1$, A$_3$ and A$_4$ characterized the N-terminal domain but, unlike Sec14, the loop connecting helices A$_1$ and A$_3$ (A$_2$ and A$_3$ in Sec14) was expanded to host a hairpin turn and another helix (A$_2$; *Figure 2C*). As typically observed in Sec14-like PITPs, the internal cavity of Sfh5 was defined by a β-sheet floor with α helices A$_5$-A$_6$ forming one side. Helix A$_7$, in conjunction with an extension loop, comprised the substructure that gates the cavity. The isostructural A$_{10}$/T$_4$ gating helices of Sfh1 and Sfh5 were positioned similarly across the opening to the internal cavity

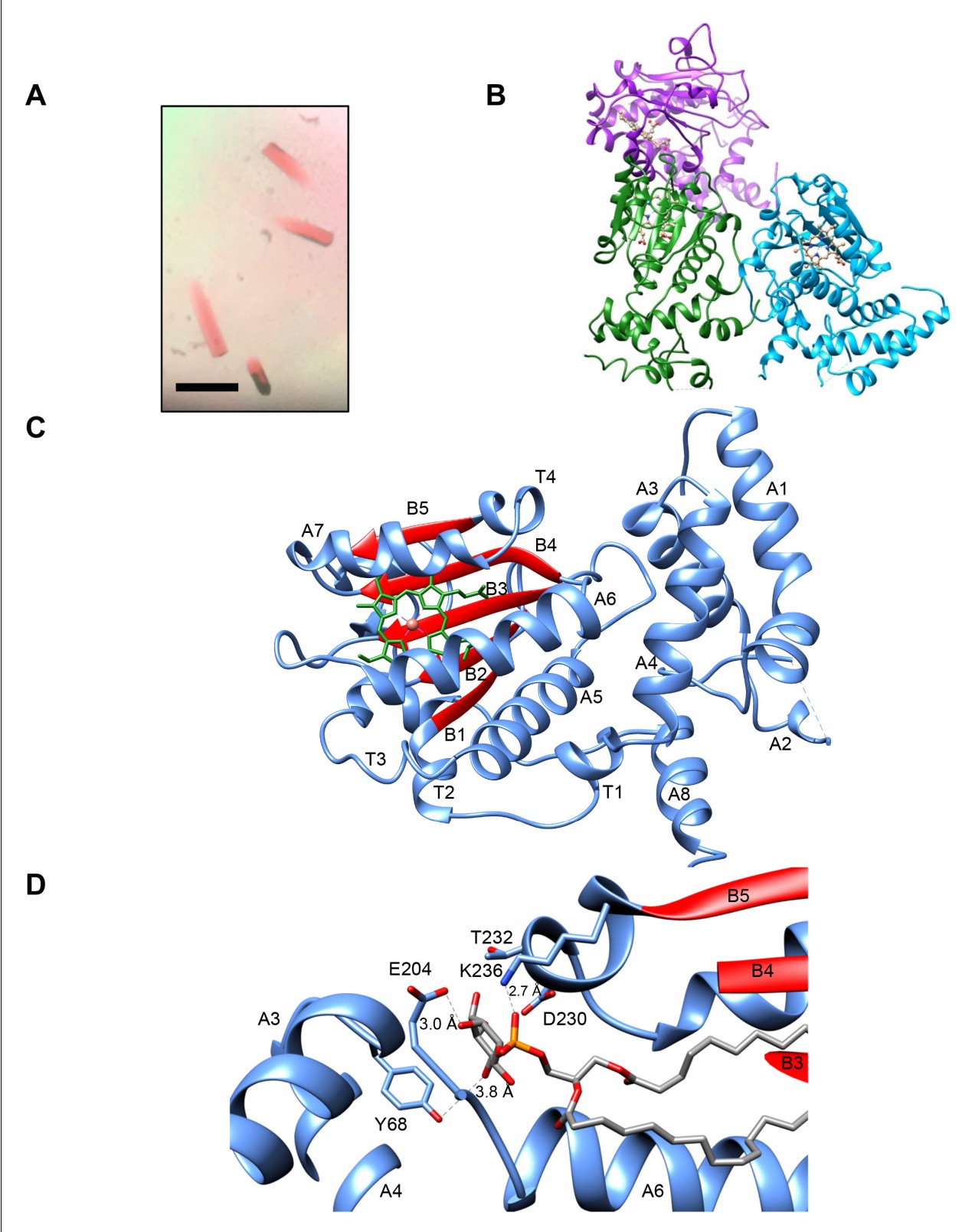

**Figure 2.** Sfh5 crystal structure. (**A**) Recombinant Sfh5 forms reddish brown rod-shaped crystals. Scale bar, 50 μm. (**B**) The asymmetric unit consists of three Sfh5 molecules and ribbon representations are colored by molecule. The bound heme is shown in ball and stick with carbon atoms colored in brown, oxygen – in red, and nitrogen in blue. (**C**) Ribbon diagram of the Sfh5 showing α-helices, loops and $3_{10}$ turns in blue, and β-strands in red. A single non-covalently-bound heme *b* molecule is rendered in green with the brown sphere representing the heme iron. (**D**) The PtdIns binding

*Figure 2 continued on next page*

*Figure 2 continued*

substructure of Sec14-like PITPs is conserved in Sfh5. A PtdIns molecule was modeled into the closed Sfh5 structure by overlaying Sfh1::PtdIns crystal structure (PDB ID: 3B7N) onto the Sfh5 crystal structure. This binding model emphasizes the conserved interactions (dashed lines) between Sec14 family PtdIns-binding barcode residues of Sfh5 and elements of the PtdIns headgroup. The corresponding distances between PtdIns headgroup structural elements and side-chains of the barcode residues in this dock model are shown.

The online version of this article includes the following figure supplement(s) for figure 2:

**Figure supplement 1.** Structural features of Sfh5.

(*Figure 2—figure supplement 1*). By contrast, the Sfh5 helical gate was shifted by ~18 Å across the cavity opening relative to the position of this substructure in Sec14 – thereby describing a 'closed' Sfh5 conformer (*Figure 2—figure supplement 1*). Again, in a strategy conserved across the Sec14-like PITP family, Sfh5 residue $F_{228}$, located at the C-terminus of the KKFV gating element (KPFL in Sec14 and Sfh1), engaged $K_{192}$ of the adjacent helix $A_6$ in a cation-π interaction to stabilize the closed conformer. Hydrophobic interactions between $V_{229}$ on the loop extending from helix $A_7$, and

**Table 1.** X-ray data collection and refinement statistics.
Parentheses indicate highest shell.

| Data Collection | Phasing data set | Refinement data set |
| --- | --- | --- |
| Wavelength (Å) | 1.55 | 1.55 |
| Space group | $P4_32_12$ | $P4_32_12$ |
| a, b, c (Å) | 205.13, 205.13, 68.27 | 205.34, 205.34, 68.31 |
| α, β, γ (°) | 90, 90, 90 | 90, 90, 90 |
| Resolution (Å) | 41–2.7 (2.8–2.7) | 29.3–2.9 (3.06–2.9) |
| $R_{merge}$ | 0.291 (1.736) | 0.157 (2.22) |
| $R_{pim}$ | 0.049 (0.545) | 0.064 (0.881) |
| $I/\sigma I$ | 13.5 (0.8) | 10.2 (1.2) |
| Completeness (%) | 98.5 (85.3) | 99.6 (100) |
| Redundancy | 32.2 (9.1) | 6.9 (7.0) |
| $CC_{1/2}$% | 0.996 (0.303) | 0.996 (0.505) |
| Refinement | | |
| Resolution (Å) | 29.3–2.9 | |
| No. of Reflections: | 61477 | |
| $R_{work}$ | 0.23 | |
| $R_{free}$ | 0.30 | |
| No. of atoms | 7169 | |
| Protein | 7040 | |
| Ligand | 129 | |
| B-factors (Å$^2$) | | |
| Protein | 79.3 | |
| Ligands | | |
| R.M.S. deviations: | 57 | |
| Bond lengths, rmsd (Å) | 0.01 | |
| Bond angles, rmsd (°) | 1.41 | |
| Ramachandran plot | | |
| Ramachandran favored (%) | 86 | |
| Ramachandran allowed (%) | 13 | |
| Ramachandran outliers (%) | 0.5 | |

$I_{195}$ of helix $A_6$, stabilized this closed conformation. Further ramifications of the $K_{192}$-$F_{228}$ interaction will be detailed below.

In Sec14 and Sfh1, the conformational dynamics of the helical gate are coupled to a switch module (G-module) that operates via rearrangement of multiple H-bonds between distinct substructures to open and close the helical gate (*Ryan et al., 2007*). A prominent component of the switch element is a string motif that wraps around the back of the protein molecule and is characterized by a series of short $3_{10}$-helices. This structural motif is conserved in Sfh5 (*Figure 2—figure supplement 1*). Although no bound phospholipid was detected in the Sfh5 ligand-binding pocket, and the pocket was occupied with heme, the structural elements that comprise the PtdIns-binding substructure in other Sec14-like PITPs are conserved in Sfh5 (*Figure 2D*).

## Coordination environment of Sfh5-bound heme

Although the final refined Sfh5 structure had poor electron densities for several unstructured regions, especially in chains A and C, the *2Fo-Fc* electron density was clear and well defined for the heme-binding region of all three chains. Bound heme was not visible in the surface render of the closed Sfh5 conformer (*Figure 3A*; left panel). The extent to which heme was buried in the 'closed' Sfh5 conformer was quantified by calculating its solvent-exposed surface area. Only ~8 $Å^2$ of the heme surface area was solvent exposed – as opposed to 835 $Å^2$ for free heme. The deep sequestration of bound heme in the Sfh5 internal cavity was visualized in a slice representation of the closed Sfh5 conformer (*Figure 3A*; right panel). The plane of the heme *b* porphyrin ring ran parallel to helix $A_6$ and to β-strands $B_3$-$B_5$ which collectively form the β-sheet floor of the binding cavity. In that configuration, heme *b* exhibited twenty-five van der Waals interactions with amino acid side chains of the residues that line the cavity, including residues $F_{138}$, $F_{144}$, $V_{178}$, $I_{187}$, $I_{195}$, $F_{211}$, $V_{212}$, $F_{218}$, $I_{225}$, $V_{229}$, and $F_{237}$ (*Figure 3B*). In addition, $F_{211}$, $Y_{128}$ and $Y_{175}$ formed π-π stacking interactions with the pyrrole groups of the heme porphyrin ring.

Heme *b* binding within the Sfh5 internal cavity exhibited two notable characteristics. First, the porphyrin conformations within the each of the three monomers (identified as chains A, B and C) were largely planar as far as could be determined at the resolution level of the structure. Second, the heme *b* coordination mechanism was unusual. Bound heme-Fe was in close proximity (2.3 Å) to the hydroxyl group of residue $Y_{175}$ (*Figure 3C*; left panel), and tyrosine coordination was consistent with the results of safranin T dye-reduction assays which estimated an unusually low thermodynamic reduction potential for Sfh5 heme (−330 ± 3 mV vs normal hydrogen electrode). The $Y_{175}$ hydroxyl group also engaged the Nε2-tautomer of the proximal $H_{173}$ via a hydrogen bond interaction (*Figure 3C*; left panel). Residue $H_{173}$ was positioned orthogonal to the heme plane and was poised to form a salt bridge between the Nδ1-tautomer and one of the heme propionate moieties. Moreover, a chain of H-bond and polar interactions was evident between the bound Fe, $Y_{175}$ and $H_{173}$, and extended from the Fe center to a propionate side chain of bound heme in contact with the porphyrin ring. The heme propionate groups also engaged in H-bonding and acid/base interactions with Sfh5 residues $Y_{128}$, $R_{148}$ and $K_{209}$. Moreover, the $S_{191}$ side-chain was in H-bond contact with the nitrogen of the heme pyrrole ring (*Figure 3C*; left panel). We note this configuration for $S_{191}$ was on display in the chain B structure whereas rotamers where the $S_{191}$ side-chain was flipped 180° away from the heme center were more consistent with the refined chain A and chain C structures. Alignment of Sfh5 homologs from multiple fungal species indicate residues $Y_{128}$, $R_{148}$, $H_{173}$, $Y_{175}$, and $K_{209}$ are invariant, and that multiple other heme-coordinating residues (including $S_{191}$) are also highly conserved – thereby defining a barcode for heme binding by Sec14-like proteins of the Sfh5 clade (*Figure 3—source data 1*). This configuration forms an extensive electronic system (*Figure 3C*; right panel).

Only two amino acid side chains ($S_{191}$ and $I_{225}$) reside within a 5 Å radius on the distal side of the Fe – reporting the absence of a sixth coordinating ligand and establishing that bound Fe is penta-coordinate. That vacant coordination site on the distal side of the Fe represents a candidate substrate binding pocket positioned close to the Sfh5 protein surface and to which access from solvent is occluded by the side chains of $K_{188}$ and $K_{192}$ of helix $A_6$ on one side and residues $L_{224}$, $I_{225}$ of helix $A_7$ and $F_{228}$ on the loop connecting helix $A_7$ and $3_{10}$-helix $T_4$, on the other (*Figure 3D*). As discussed above, $K_{192}$ and $F_{228}$ engage in a cation-π interaction that stabilizes the closed conformation of the Sfh5 helical gate whose isostructural elements control access to the lipid binding cavity in Sec14 and other Sec14-like PITPs. This arrangement suggests a mechanism where rotamer rearrangements of

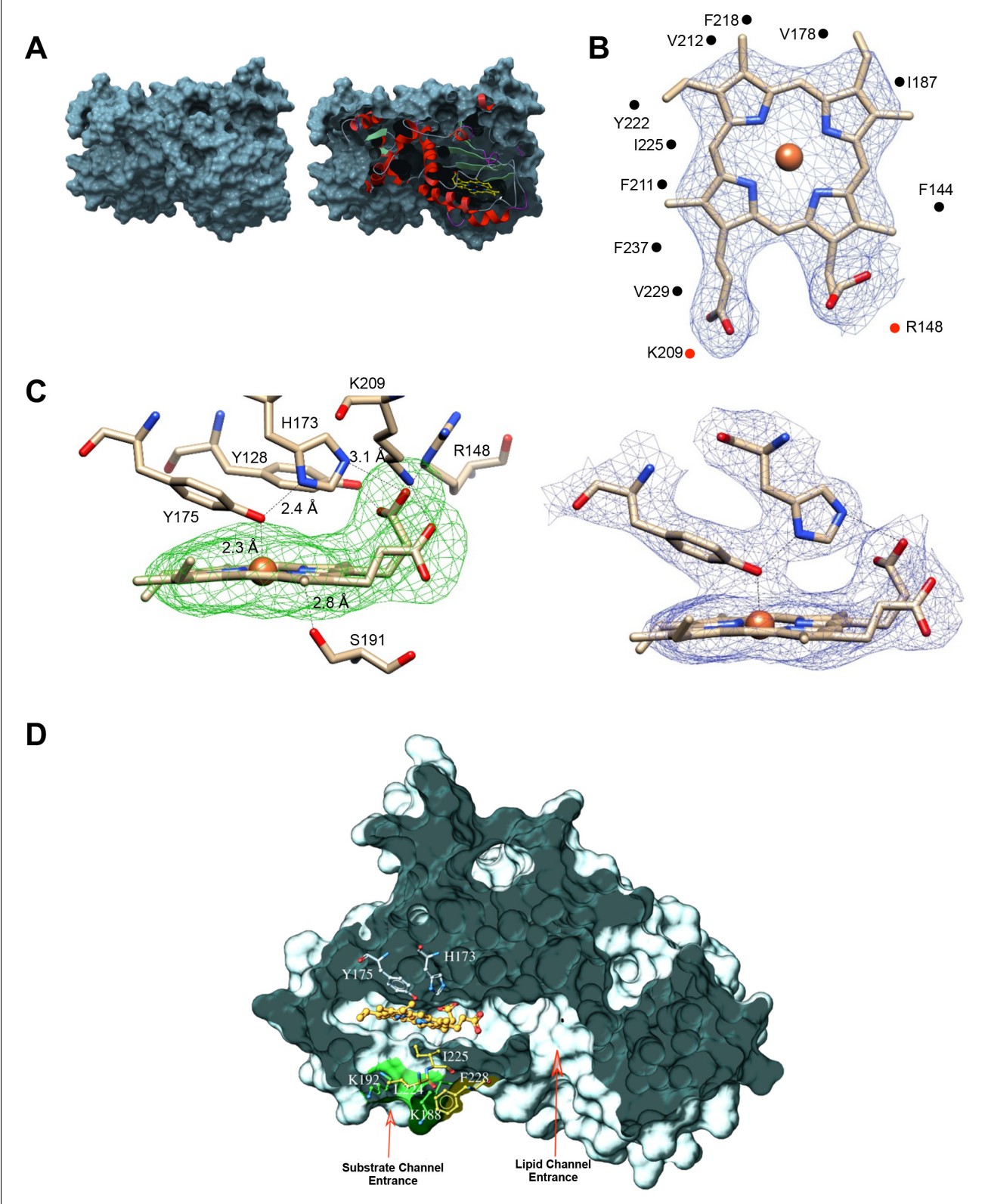

**Figure 3.** Coordination environment of Sfh5-bound heme. (**A**) Left panel shows the surface rendering of Sfh5 in grey. Right panel shows surface view of Sfh5 clipped to expose the buried heme rendered in yellow ball and stick with protein secondary structure elements depicted in cartoon ribbon. (**B**) Heme bound to chain B of the Sfh5 crystal structure. The 2Fo-Fc electron density map for heme contoured at 1.5 σ is displayed as blue mesh. Residues engaged in van der Waals contacts with heme are shown as gray spheres. The side-chains of residues interacting with the propionyl groups of heme

*Figure 3 continued on next page*

*Figure 3 continued*

(K$_{209}$ and R$_{148}$) are presented as red dots. (**C**) Left panel: Sfh5 residues that bind heme are shown including the Fe-coordinating residues Y$_{175}$ and H$_{173}$. Dashed lines identify the inter-atomic distances between the indicated Sfh5 residues and cognate components of heme. Polder omit electron density map for heme is shown as green mesh (*Liebschner et al., 2017*), contoured at 2.5 σ. Right panel: Lateral view of the 2Fo-Fc map showing electron density associated with residues coordinating the heme iron center (contoured at 1.5σ). (**D**) Surface view of Sfh5 clipped to display the ligand-binding cavity. Bound heme (shown in ball and stick and colored by the element with carbons in gold) resides deep inside the cavity. Heme-coordinating residues H$_{173}$ and Y$_{175}$ are displayed in ball and stick mode and colored by element with carbons in light gray. Indicated access to the vacant heme coordination site from solvent is potentially controlled by conformational transitions of the K$_{188}$ and K$_{192}$ side chains (labeled as substrate channel, entrance displayed in ball and stick render and colored by element with carbons in green), and the side-chains of residues L$_{224}$, I$_{225}$ and F$_{228}$ (colored by element with carbons in yellow). The surface contributed by the side chains of residues K$_{188}$ and K$_{192}$ is colored in green. Access channel for lipid to the internal cavity of other Sec14-like PITPs is also indicated.

The online version of this article includes the following source data for figure 3:

**Source data 1.** Fungal Sfh5 orthologs.

these side chains, and/or conformational transitions of the A$_7$ helix, gate substrate access into the vacant heme coordination site. Notably, K$_{188}$ and K$_{192}$ are absolutely conserved across the Sfh5 clade, whereas the F$_{228}$ is highly conserved. In cases where the F$_{228}$ cognate is a valine, it is the K$_{188}$ cognate that is projected to engage in a cation-π interaction with a phenylalanine that replaces the cognate L$_{224}$ (*Figure 3—source data 1*).

## Roles of residues Y$_{175}$ and H$_{173}$ in heme binding and Fe-center chemistry

Heme-binding tyrosines can be stabilized by adjacent ionizable residues that enhance the tyrosinate character and strengthen the Tyr O-Fe bond (*Pluym et al., 2008*; *Sharp et al., 2007*; *Caillet-Saguy et al., 2008*). The Sfh5 structure suggested that the H$_{173}$ Nδ1 proton acts as an H-bond donor to Y$_{175}$ – thus transferring negative charge from the tyrosinate ion to the H$_{173}$ imidazole ring and modulating strength of the Y$_{175}$-Fe interaction. To test this possibility, a series of Sfh5 Y$_{175}$ and H$_{173}$ missense mutants was generated and the consequences on heme binding characterized. As demonstrated in other experiments described below, the relevant Sfh5 heme-binding mutants were all properly folded proteins.

That heme binding was compromised in specific mutants was readily apparent from the colors of the purified recombinant proteins (*Figure 4A*). Such deficits were quantified by the decline in Soret peak intensities and heme content as determined by pyridine hemochromagen assays and ICP-MS analyses of the mutant proteins (*Figure 4B*, *Supplementary file 1*). Substitution of the axial Y$_{175}$ by non-polar residues (i.e. Sfh5$^{Y175A}$ and Sfh$^{Y175F}$) completely abrogated heme binding whereas the Sfh5$^{Y175H}$ mutant largely retained the ability to bind heme – revealing the necessity of a polar axial ligand for efficient heme binding (*Supplementary file 1*). The Sfh5$^{H173A}$ variant showed an ~50% reduction in heme binding, while the Sfh5$^{H173Y}$ variant, unlike Sfh5$^{H173A}$, purified with heme loaded at levels comparable to Sfh5 – further highlighting the role of the amino acid co-axial to residue Y$_{175}$ in stabilizing the critical Y$_{175}$-Fe interaction. The Soret maxima for Sfh5$^{H173Y}$ shifted to 402 nm as compared to 404 nm for wild-type Sfh5 with no appreciable change upon exposure to excess dithionite (*Figure 4C*). By contrast, the Soret maximum of Sfh5$^{Y175H}$ was centered at 407 nm and red-shifted to 430 nm upon reduction of the protein with dithionite – suggesting the heme in this mutant was in the oxidized Fe$^{3+}$ state (*Figure 4D*). These results demonstrated that the Y$_{175}$ interaction with Fe was essential for stabilizing heme binding, and that H$_{173}$ played an important role in enhancing Lewis basicity of the Y$_{175}$ hydroxyl in its interaction with heme.

## Biophysical properties of the Sfh5 Fe-center

The 10 K X-band electron paramagnetic resonance (EPR) spectrum of oxidized wild-type Sfh5 exhibited an axial signal with $g_\perp$=5.83 and $g_\parallel$=2.00. This spectrum is typical of high-spin S = 5/2 Fe$^{3+}$ heme centers with the rhombicity parameter E/D ≈ 0 (*Figure 5A*; *García-Rubio et al., 2007*; *Adachi et al., 1993*). No signals reflecting low-spin Fe$^{3+}$ hemes were detected. The Fe$^{3+}$ heme assignment was further supported by the loss of the EPR signal upon reduction with dithionite. Taken together, these data confirm that the heme prosthetic group is redox-active and that the

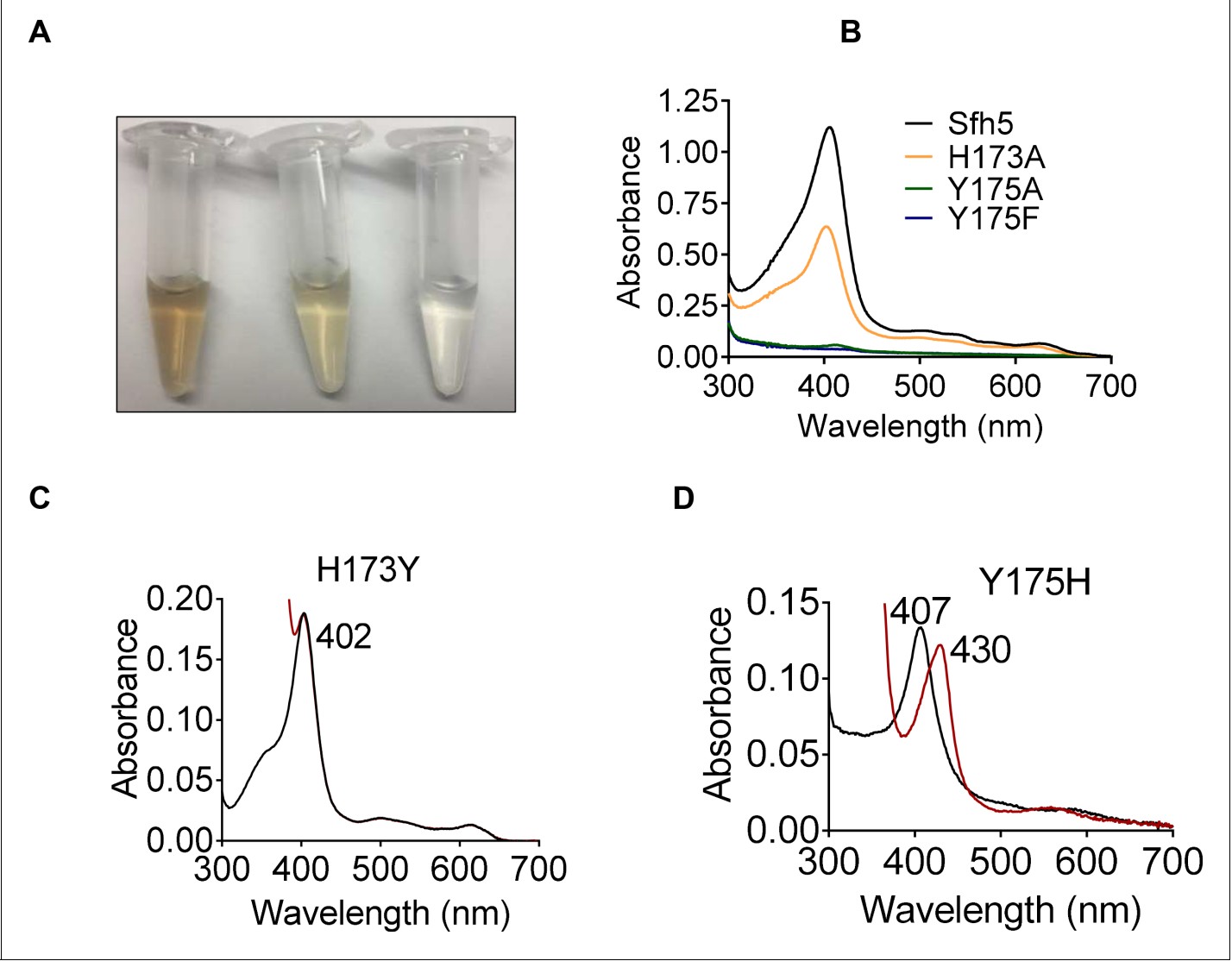

**Figure 4.** Roles of Sfh5 residues $Y_{175}$ and $H_{173}$ in heme binding and Fe-center chemistry. (**A**) The color intensities of solutions (30 μM) of purified recombinant Sfh5 (left), Sfh5$^{H173A}$ (middle) and Sfh5$^{Y175F}$ (right) are compared. (**B**) UV-vis absorption spectra of Sfh5, Sfh5$^{H173A}$, Sfh5$^{Y175A}$ and Sfh5$^{Y175F}$ are shown. The protein concentrations were fixed at 30 μM. (**C, D**) UV-vis spectra of Sfh5$^{H173Y}$ and Sfh5$^{Y175H}$ are shown, respectively. Spectra obtained after reduction of sample with dithionite are in red. Sfh5$^{H173Y}$ did not undergo a spectral shift upon reduction.

oxidized $Fe^{3+}$ state is a high-spin penta-coordinate with a vacant axial site. Moreover, the data show that the reduced state is EPR-silent – consistent with reduction to the $Fe^{2+}$ state.

The ability of the vacant coordination site to bind diatomic molecules was interrogated. In one set of experiments, exposure of oxidized $Fe^{3+}$ Sfh5 to a molar excess of potassium cyanide (KCN) induced a marked red-shift of the Soret peak from 404 to 417 nm (**Figure 5B**). Heme titration experiments with CN⁻ reported a 1:1 binding stoichiometry – indicating one bound CN⁻ per $Fe^{3+}$ heme center (**Figure 5—figure supplement 1**). Cyanide ions did not appear to coordinate the reduced heme center as no shift in the Soret band was observed when Sfh5 was first reduced with excess sodium dithionite prior to exposure to CN⁻ (**Figure 5C**).

In a second set of experiments, oxidized Sfh5 was exposed to a CO atmosphere. This exposure failed to induce a spectral shift in the Soret profile (**Figure 5D**) – suggesting that CO did not bind to Sfh5 $Fe^{3+}$ heme. However, exposure of dithionite-reduced Sfh5 to CO resulted in a blue shift of the Soret maximum from 426 to 416 nm (**Figure 5E**) – demonstrating CO binding to reduced Sfh5 and suggesting formation of a low-spin $Fe^{2+}$ hemoprotein carbonyl complex. Analyses of the α-bands

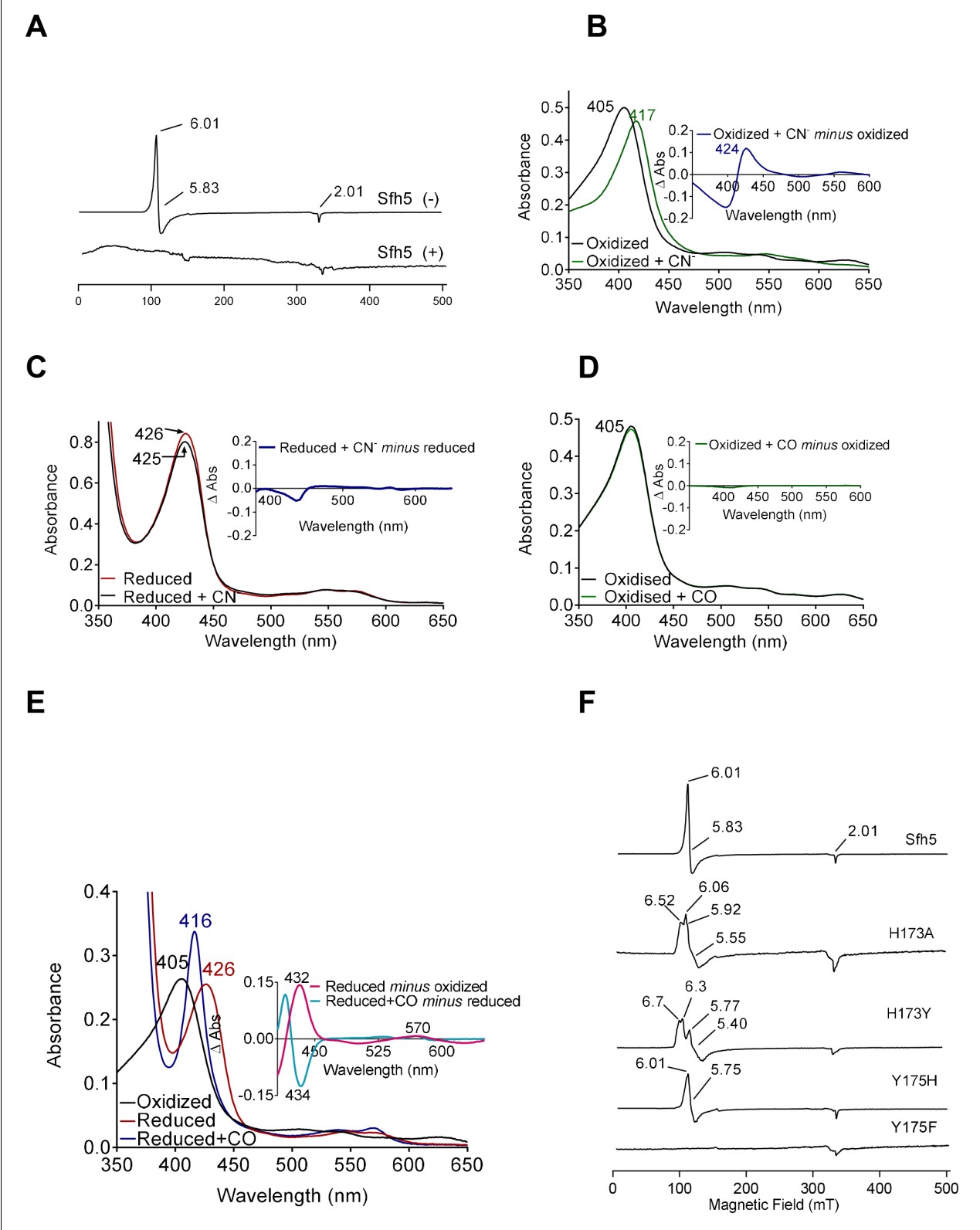

**Figure 5.** Electronic properties of the Sfh5 Fe-center. (**A**) EPR spectra of purified Sfh5 before (-) and after dithionite treatment (+). (**B**) UV-visible absorption spectrum of purified (oxidized) Sfh5 without and with treatment with potassium ferricyanide as source of cyanide ion (CN⁻) (***Blumenthal and Kassner, 1980***). Inset shows the difference spectrum between the two conditions and highlights a significant red shift in the Soret maximum induced by CN⁻. (**C**) Purified Sfh5 was first reduced with excess dithionite and spectra were taken before and after incubation in the presence of CN⁻. Inset

*Figure 5 continued on next page*

*Figure 5 continued*

indicates the difference spectrum between the two conditions and shows no appreciable shift in the Soret maximum. (D) UV-vis spectra before and after incubation of purified Sfh5 with carbon monoxide (CO) gas are shown. Inset indicates the difference spectrum between the two conditions and shows no shift in the Soret maximum in the presence of CO. (E) UV-vis spectra of purified Sfh5 reduced with dithionite and incubated in the presence of CO. Inset, difference spectra as indicated. (F) EPR spectra of purified Sfh5, Sfh5$^{H173A}$, Sfh5$^{H173Y}$, Sfh5$^{Y175H}$, and Sfh5$^{Y175F}$ are shown. Protein samples were normalized by concentration (150 μM).

The online version of this article includes the following figure supplement(s) for figure 5:

**Figure supplement 1.** Accessibility of the open axial binding site of Sfh5 by CN⁻.

under those conditions indicated CO bound stoichiometrically to reduced heme (100% of reduced Sfh5 bound CO; *Figure 5E*, inset). The collective results indicate that the vacant axial site of Sfh5 heme can coordinate small exogenous ligands with CN⁻ and CO binding the $Fe^{3+}$ and $Fe^{2+}$ state, respectively.

The effects of $Y_{175}$ and $H_{173}$ on Sfh5 heme electronics were also assessed. The EPR signal exhibited by the Sfh5$^{H173A}$ mutant showed features similar to those of wild-type Sfh5, suggesting that the mutant heme center was also high spin $Fe^{3+}$, albeit with a rhombic parameter (E/D ≈ 0.02) that is more typical of tyrosine-ligated five coordinate (5 c) hemes (*Adachi et al., 1991*). A rhombic g-tensor was evident in the spectrum of Sfh5$^{H173Y}$, whereas Sfh5$^{Y175H}$ presented spectral features most similar to those of wild-type Sfh5 (*Figure 5F*). Taken together, these data indicate that the Fe-tyrosine bond is subject to modulation by the co-axial $H_{173}$.

As is the case with other heme-containing proteins, Sfh5 exhibited pseudo-peroxidase activity when pyrogallol was offered as a substrate and its oxidation to purpugallin was followed (see Materials and Methods). The observed low-level activity (relative to a horseradish peroxidase control) was abolished in Sfh5$^{H173A}$, but was stimulated 4-fold in Sfh5$^{Y175H}$ (*Supplementary file 1*). While catalases are heme proteins with tyrosine axial ligands, purified Sfh5 exhibited no measurable catalase activity in vitro.

## Sfh5 heme center exhibits two magnetic states

The electronic properties of the Sfh5 heme center were investigated in greater detail using Mössbauer spectroscopy. Simulation of low-temperature, low-field (5 K, 0.05 T) spectra of oxidized $^{57}$Fe-enriched wild-type Sfh5 (*Figure 6A*) required consideration of two magnetic states. The solid red line describes a simulation assuming two S = 5/2 Fe$^{III}$ heme states that we term LA (Large A) and SA (small A). The LA state exhibited larger average hyperfine couplings (A-values) whereas the SA state was characterized by smaller A-values (*Supplementary file 1*). The fitting parameters were similar to those previously obtained for high-spin $Fe^{3+}$ hemes with phenolate coordination (*Bominaar et al., 1992*).

The six independently generated Sfh5 preparations analyzed exhibited similar spectra but required somewhat different relative intensity ratios for the two magnetic states from preparation to preparation. The high-field (6 T) spectrum of one $^{57}$Fe-enriched Sfh5 preparation could also be fitted using the same two-state model (*Figure 6—figure supplement 1*). The magnetic hyperfine interactions associated with the LA state were partially collapsed by warming the sample to 150 K whereas those for the SA state were not (data not shown). The Mössbauer spectrum of another Sfh5 preparation was dominated by the SA state (*Figure 6B*). After treatment with dithionite, the $^{57}$Fe-enriched Sfh5 exhibited a quadrupole doublet typical of high-spin $Fe^{2+}$ (*Figure 6C*), confirming the $Fe^{3+/2+}$ redox activity of Sfh5 heme. By contrast, simulating the Mössbauer spectrum of $^{57}$Fe-enriched Sfh5$^{H173A}$ required only consideration of the LA state (*Figure 6D*). Moreover, the collapsed LA state recorded for $^{57}$Fe-enriched Sfh5 fit to a quadrupole doublet with the same isomer shift δ and quadrupole splitting $\Delta E_Q$ used to simulate the Sfh5$^{H173A}$ spectrum. These findings further reinforce the case for a two magnetic state model for the Sfh5 heme center.

## Incompatibility of high affinity Sfh5 heme-binding with PtdIns binding/ exchange

A distinguishing property of Sfh5 is that, among the yeast Sec14-like PITPs, this protein exhibits only marginal PtdIns-exchange activity in vitro and is incapable of significantly stimulating PtdIns 4-OH

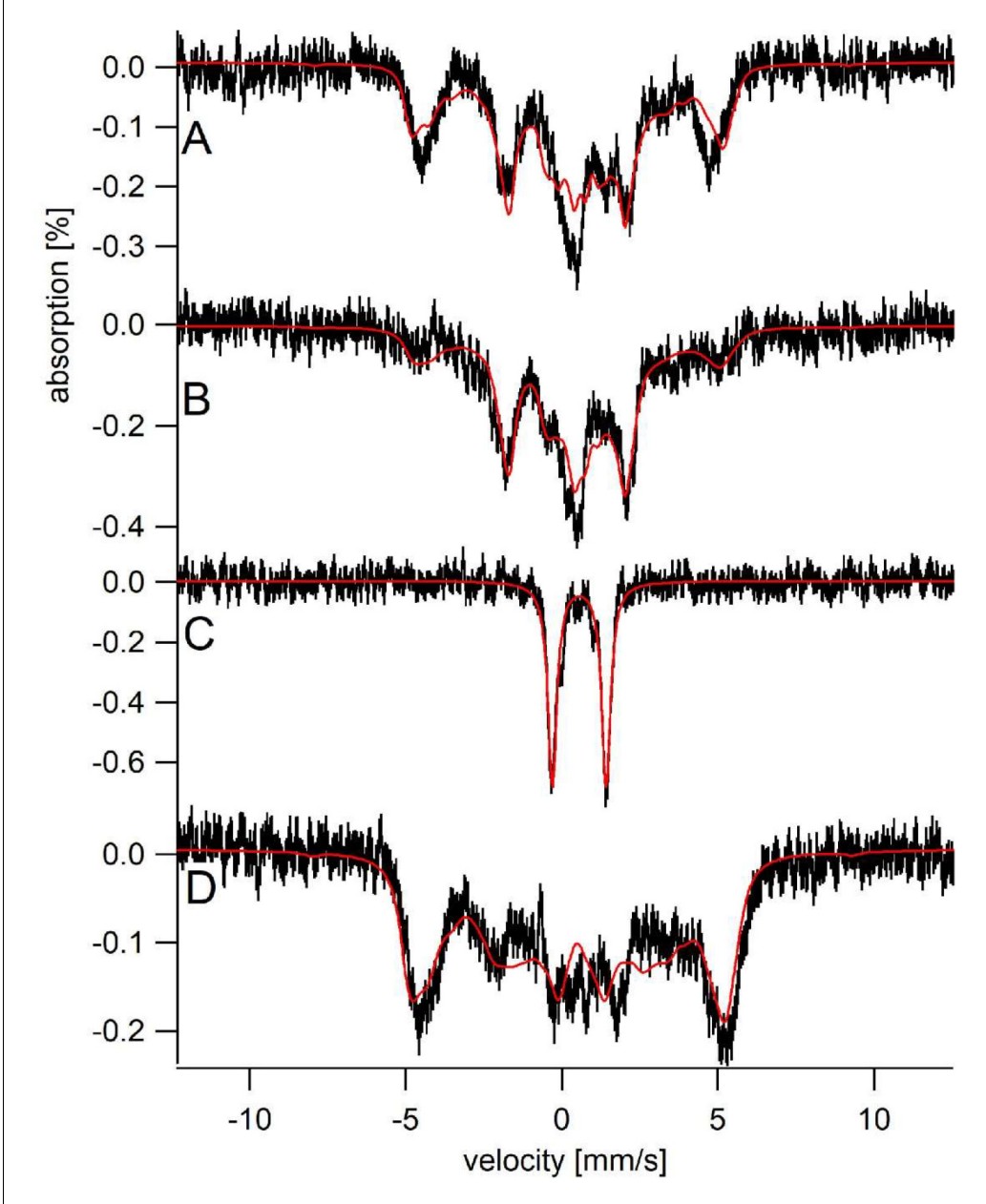

**Figure 6.** Sfh5 heme center exhibits two magnetic states. Mössbauer spectra were collected at low-temperature (5 K) and field (0.05 T). (**A**) Spectrum of Sfh5 sample prepared at pH 7. The red line is a composite simulation with 63% $Fe^{III}_{LA}$ site and 37% $Fe^{III}_{SA}$ site. (**B**) Spectrum of an independent sample of Sfh5 prepared at pH 8. (**C**) Spectrum of the sample of Sfh5 in (**B**) reduced with sodium dithionite. The simulation in B was used to remove 60% of the overall spectral intensity in C. (**D**) Spectrum of Sfh5$^{H173A}$100% $Fe^{III}_{LA}$ site. In all spectra, the field was applied parallel to the gamma radiation.

The online version of this article includes the following figure supplement(s) for figure 6:

**Figure supplement 1.** High-field spectrum of $^{57}$Fe-enriched Sfh5.

kinase activity in vivo – as evidenced by the fact that its high-level expression is unable to rescue growth of cells depleted of Sec14 activity (*Li et al., 2000*). These properties are evident even though Sfh5 conserves the PtdIns-binding barcode that is the cardinal signature of the Sec14 superfamily (*Schaaf et al., 2008*; *Bankaitis et al., 2010*). We therefore considered the possibility that Sfh5 does not function as a PITP per se, and that heme is not an exchangeable second ligand. This property

would be unique to Sfh5 as other members of the Sec14-like PITP family all have the capacity to host an exchangeable second lipid (*Tripathi et al., 2019*).

Based on the fact that Sfh5 exhibits an intact PtdIns-binding barcode, we hypothesized that abrogation of heme-binding might reactivate PtdIns- exchange activity in Sfh5. This prediction was independently examined using both in vitro and in vivo approaches. First, the PtdIns-exchange activities of purified Sfh5 and Sfh5 heme-binding mutants were tested using an assay that monitors transfer of [$^3$H]-PtdIns from rat liver microsomes to PtdCho liposomes in vitro. Whereas wild-type Sfh5 showed little activity, as previously reported (*Li et al., 2000*), the heme-less Sfh5$^{Y175F}$ variant exhibited enhanced activity relative to wild-type – the specific activity of Sfh5$^{Y175F}$ was ~70% of Sec14 relative to ~10% for Sfh5. The Sfh5$^{H173A}$ mutant with reduced heme-binding capacity also showed enhanced activity – one that was intermediate between those of wild-type Sfh5 and Sfh5$^{Y175F}$ (*Figure 7A*). These data indicated that Sfh5 PtdIns-transfer activity is inversely proportional to heme-binding affinity, and that these heme-binding mutants are well-folded proteins in functional conformations. As expected, the purified Sfh5$^{Y68A,Y175F}$ and Sfh5$^{Y175,E204VF}$ variants with compromised PtdIns-binding barcodes were significantly defective in PtdIns-transfer activity in vitro relative to the Sfh5$^{Y175F}$ control.

Second, the in vitro data were reinforced by two sets of in vivo results. Whereas high-level expression of wild-type Sfh5 was unable to rescue *sec14-1$^{ts}$* associated growth defects when the test strain was challenged at the restrictive temperature of 37°C, Sfh5$^{Y175F}$ expression supported a robust rescue of the growth defect (*Figure 7B*, left panel). Similar data were forthcoming from plasmid-shuffle assays where the abilities of Sfh5 and its variants to bypass the essential cellular requirement for Sec14 activity, when expressed at high levels, were examined (*Figure 7B*, right panel). Whereas high-level expression of Sfh5$^{Y175F}$ was capable of supporting viability of *sec14Δ* segregants, expression of wild-type Sfh5 was not able to do so.

As ergosterol synthesis is the sole 'essential' heme-dependent activity in yeast grown with glucose as sole carbon source, we tested whether resuscitation of PtdIns-exchange activity of wild-type Sfh5 was recapitulated when Sfh5 was expressed in heme-less *hem1Δ* yeast mutants cultured in the presence of ergosterol. Unlike the case for the Sfh5$^{Y175}$ mutant, enhanced expression of WT Sfh5 was unable to rescue growth of the in *hem1Δ sec14-1$^{ts}$* mutant at 37 °C (*Figure 7C*). That is, wild-type Sfh5 produced in a heme-free cellular context still failed to exhibit PITP activity. Taken in aggregate, these data report that Sfh5 does not function as a PtdIns/heme-exchange protein in vitro or in vivo.

Another unforeseen result came from reciprocal experiments where PtdIns-binding deficits were introduced into Sfh5 mutants strongly defective in heme-binding. In those experiments, the Y$_{68}$A substitution that compromises the Sfh5 PtdIns-binding barcode was incorporated into the heme-less Sfh5$^{Y175F}$ context. Surprisingly, the Sfh5$^{Y68A,Y175F}$ double mutant protein showed unambiguous resuscitation of low level heme binding. That is, Sfh5$^{Y68A,Y175F}$ showed a weak, yet distinct, Soret peak at 413 nm (*Figure 7—figure supplement 1*), and a weak EPR signature of high spin Fe$^{3+}$ heme, while Sfh5$^{Y175F}$ did not (*Figure 7—figure supplement 1*). Thus, alterations in the Sfh5 PtdIns-binding substructure defects can partially relieve heme-binding defects – even though the structural elements that coordinate PtdIns headgroup binding and heme-binding are physically distinct. There is specificity to this effect as the E$_{204}$V PtdIns-binding barcode mutation, which effectively abolishes PtdIns-binding by Sfh5, did not resuscitate measurable heme-binding in the Sfh5$^{Y175F}$ context (data not shown).

## Sfh5 heme is not exchangeable with an apo-myoglobin acceptor

As Sfh5 exhibits feeble PtdIns/heme exchange activity in vitro, and it does not exhibit the properties of a canonical PITP in vivo, we considered the possibility that Sfh5 donates bound heme to a suitable acceptor apo-protein. To test this hypothesis, an in vitro assay was developed where Sfh5 served as heme donor and apo-myoglobin was incorporated as an avid heme acceptor. We took advantage of the pseudo-peroxidase activities of myoglobin and of heme-bound Sfh5 to monitor transfer of heme from Sfh5 to apo-myoglobin. Although apo-myoglobin readily bound free heme presented as hemin, no transfer of heme between Sfh5 donor and apo-myoglobin acceptor was recorded in this assay regime – even after prolonged incubation (*Figure 8*). Reduction of Sfh5 heme with dithionite was ineffective in promoting heme transfer to apo-myoglobin (data not shown). These data indicate that

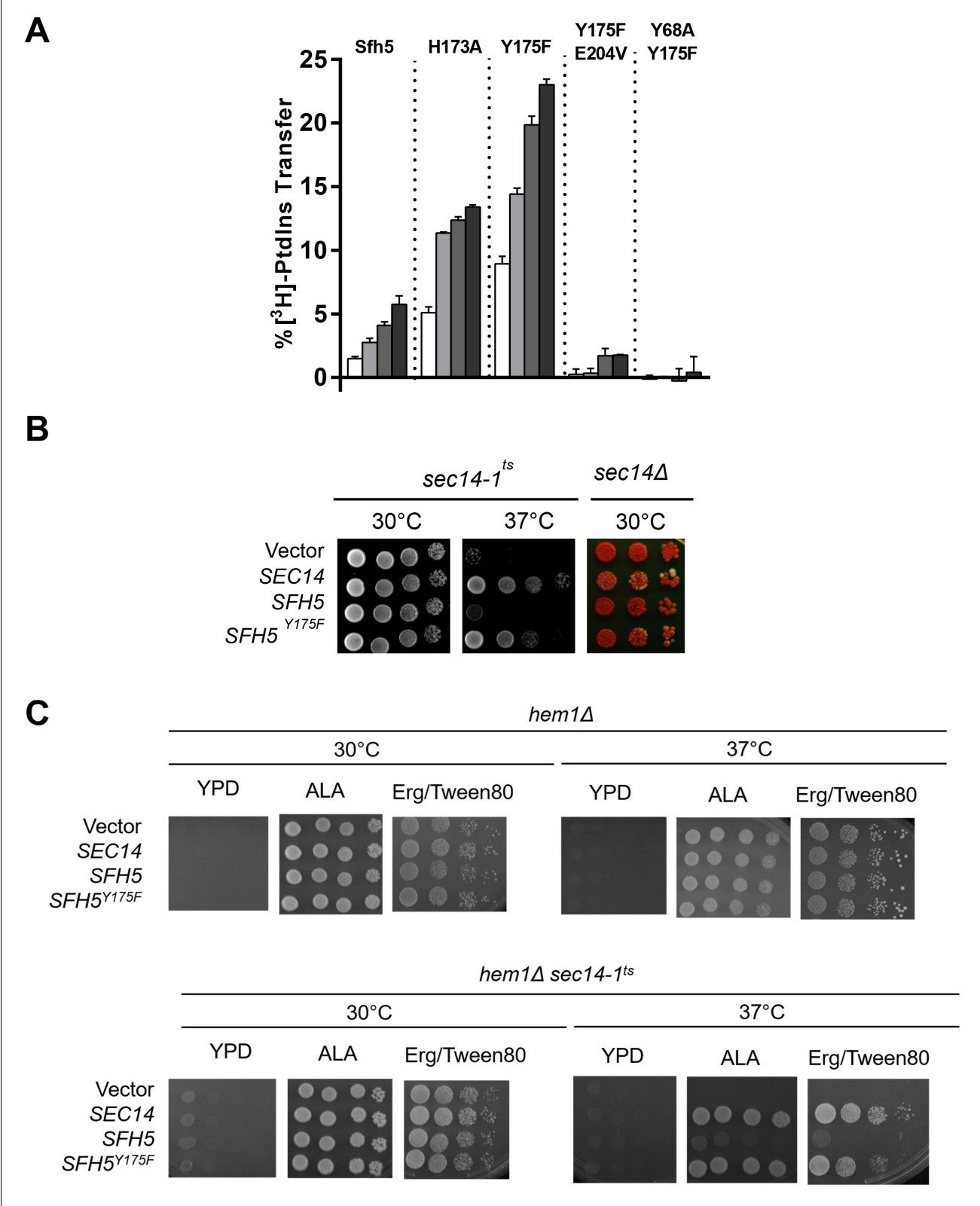

**Figure 7.** Incompatibility of high affinity Sfh5 heme-binding with PtdIns binding. (**A**) In vitro [³H]-PtdIns transfer assays were performed with purified recombinant Sfh5, Sfh5[H173A] and Sfh5[Y175F] in a titration series where protein inputs were increased in steps of two-fold (5, 10, 20, 40 µg). The total radiolabeled input for each assay varied between 8377 to 12,070 cpm, and background ranged between 150–575 cpm. The transfer values represent the mean of triplicate assay determinations from at least two independent experiments. (**B**) Resuscitation of PITP activity in heme-deficient Sfh5 mutants

*Figure 7 continued on next page*

*Figure 7 continued*

in vivo. Left panel: A *sec14-1*$^{ts}$ strain was transformed with episomal YEp(*URA3*) plasmids that drive constitutive ectopic expression of *SEC14*, *SFH5* or *sfh5*$^{Y175F}$ from the powerful *PMA1* promoter. The transformants were spotted in 10-fold dilution series onto YPD plates and incubated at the indicated temperatures for 48 hr before imaging. Growth at 37°C reports rescue of growth defects associated with the *sec14-1*$^{ts}$ allele at this normally restrictive temperature. Right panel: A *sec14Δ ade2 ade3*/YEp(*SEC14, LEU2, ADE3*) strain was transformed with the indicated YEp(*URA3*) expression plasmids described in (**B**). Transformants were again spotted in dilution series onto YPD plate as above. Under those conditions, all nutrient selections are relieved and loss of the normally essential YEp(*SEC14, LEU2, ADE3*) plasmid (which has the dual properties of covering the lethal *sec14Δ* allele and also the *ade3* allele which coverts colony color of *ade2* cells from red to white) can be monitored. Appearance of white colonies that are phenotypically Leu auxotrophs signifies loss of the parental YEp(*SEC14, LEU2, ADE3*) plasmid on the basis of the YEp(*URA3*) plasmid driving expression of a functional PITP. Uniformly red colonies report the YEp(*URA3*) expression plasmid does not drive production of a functional PITP with the capacity to provide Sec14-like functions to Sec14-deficient cells. YEp(*SEC14*) serves as positive control in these experiments. (**C**) PITP activity is not resuscitated in Sfh5 expressed in yeast cells devoid of heme. Isogenic *hem1Δ* and *hem1Δ sec14-1*$^{ts}$ strains deficient in α-aminolevulenic acid (ALA) synthesis were transformed with YEp(*URA3*) plasmid as mock control or for expression of *SEC14*, *SFH5* or *sfh5*$^{Y175F}$ as indicated. Transformants were selected on a minimal media plate without uracil and supplemented with 250 µM ALA. After depleting cells for residual heme by growth in ergosterol-containing medium (added to a 20 mg/l final concentration from a 0.2% stock solution in 1:1 ethanol: Tween 80), cells were spotted onto YPD solid medium or onto YPD solid medium supplemented with ALA (250 µM) to rescue the *hem1Δ* deficiency, or onto YPD solid medium supplemented with 50 µM ergosterol/Tween 80 (Erg/Tween80) to rescue the heme deficiency without permitting heme synthesis. The cells were incubated for 72 hr at the indicated temperatures prior to imaging. The *hem1Δ* control cells grow under all conditions that either rescue heme synthesis (+ALA) or provide exogenous ergosterol whose synthesis is the sole essential heme-dependent activity in cells cultured under these conditions. Growth of the *hem1Δ sec14-1*$^{ts}$ mutant at the restrictive temperature of 37C was not rescued by enhanced expression of WT Sfh5 under heme-less conditions (i.e. on YPD supplemented with Erg/Tween80). Expression of *SEC14* or the *sfh5*$^{Y175F}$ heme-binding mutant served as additional positive controls.

The online version of this article includes the following figure supplement(s) for figure 7:

**Figure supplement 1.** Partial restoration of heme binding to Sfh5$^{Y175F}$ by alteration of a residue of the PtdIns-binding barcode.

Sfh5 binds heme with high affinity, and lend further support to the concept that Sfh5-bound heme is not an exchangeable ligand.

## Sensitivity of Sfh5-deficient cells to an organic oxidant

The biological function of Sfh5 is enigmatic as it is expressed at very low levels, the subcellular localization of the endogenous protein is unknown, and its function is non-essential for yeast viability (*Li et al., 2000*). Indeed, there is unusually little information regarding its function that can be gleaned from genetic or physical interaction databases. Consistent with that, *sfh5Δ* mutants were not compromised for growth on non-fermentable carbon sources (*Figure 8—figure supplement 1*) – indicating Sfh5 does not play a major cellular role in either regulating heme homeostasis or in loading of apo-proteins with heme. To assess whether Sfh5 plays some role in either Fe or heme uptake, *sfh5Δ* strains were cultured in media containing the Fe-chelator bathophenanthroline to induce an Fe-limiting environment. Neither functional ablation nor overexpression of Sfh5 had any significant effect on cell growth under those conditions (*Figure 8—figure supplement 1*). To investigate a hemin uptake activity for Sfh5, we took advantage of a *hem1Δ* (δ-aminolevulenate synthase-deficient) yeast mutant unable to synthesize heme. This mutant requires exogenous δ-aminolevulenic acid for survival and, as hemin is not efficiently transported into yeast cells, low concentrations of hemin fail to restore viability to *hem1Δ* cells (*Yuan et al., 2012*). Whereas expression of the *C. elegans* heme transporter Hrg4 in *hem1Δ* yeast rescued growth in media supplemented with low concentrations of hemin, overexpression of Sfh5 did not – contraindicating a heme-transport function for Sfh5 (*Figure 8—figure supplement 1*).

The collective data suggest that Sfh5 function is not essential under the conditions typically interrogated in the laboratory. However, mining of a comprehensive HIPHOP chemogenomics database identified a significant and specific sensitivity of Sfh5-deficient yeast cells to challenge with the synthetic organosulfur dithiolethione oltipraz. That phenotype was interrogated in the context of a diploid library of yeast deletion mutants where each essential yeast gene was represented in a heterozygous +/Δ arrangement while each non-essential gene was represented in a homozygous Δ/Δ arrangement (*Lee et al., 2014*). Oltipraz induces superoxide radical formation and consequently triggers a general 'heme-requiring' response in yeast (*Kang et al., 2012*; *Velayutham et al., 2007*). This signature is manifested by the fact that heterozygous diploid yeast strains reduced for heme-biosynthetic capacity (*+/hem1Δ*, *+/hem2Δ*, *+/hem3Δ*, and *+/hem12Δ*) score as having an enhanced sensitivity to oltipraz (*Figure 8—figure supplement 2*).

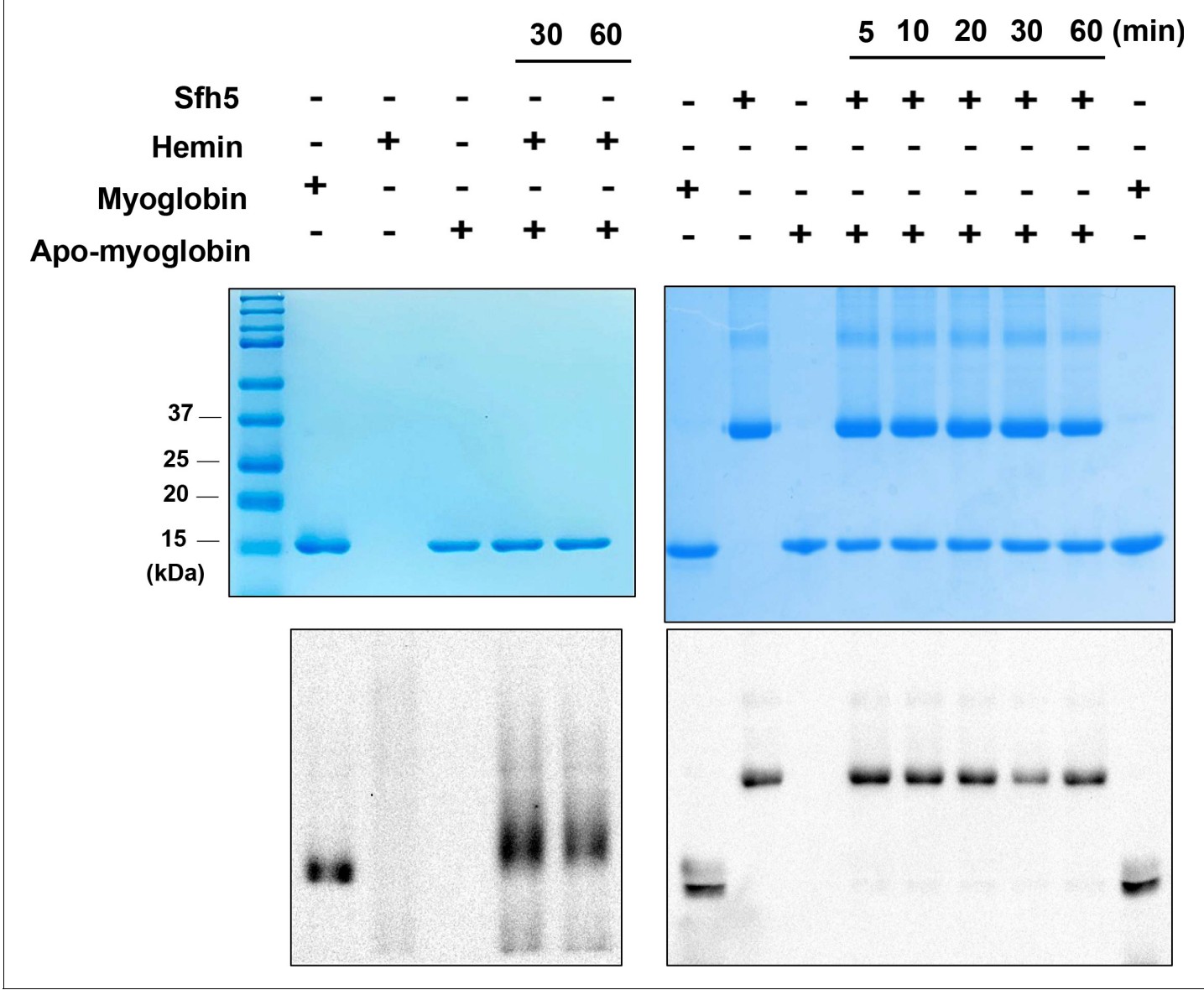

**Figure 8.** Sfh5 does not donate bound heme to an efficient heme scavenger. Top panels: Coomassie-stained SDS-PAGE gel images of Sfh5, myoglobin and apo-myoglobin proteins run individually and mixtures of hemin/apo-myoglobin and Sfh5/apo-myoglobin (as indicated) that were pre-incubated for the indicated times. The proteins migrate with the expected masses of 35.7 kDa for the Sfh5 monomer and 18 kDa for myoglobin/apo-myoglobin. Bottom panels show corresponding chemiluminescent images of the nitrocellulose membranes onto which the proteins from duplicate SDS-PAGE gels were transferred and probed by visualizing pseudo-peroxidase activity in situ. Whereas apo-myoglobin avidly scavenged hemin from the medium, no measurable transfer of heme from Sfh5 to apo-myoglobin was detected.

The online version of this article includes the following figure supplement(s) for figure 8:

**Figure supplement 1.** Sfh5 deficient cells are not compromised for major heme-requiring functions.

**Figure supplement 2.** Specific sensitivity of Sfh5 deficient cells to oltipraz.

When nonessential genes are analyzed in homozygous null mutant backgrounds, mutants with diminished capacities for scavenging reactive oxygen species (e.g. mutants functionally ablated for Sod1 superoxide dismutase activity and the Sod1 copper chaperone Ccs1) also score as hypersensitive to oltipraz – as do *sfh5Δ* cells (*Figure 8—figure supplement 2*). None of the other five yeast Sec14-like PITPs are included in this chemogenomic signature, although enrichment of mutants defective in lobe B elements of the COG complex that regulates protein trafficking through the Golgi complex is notable. When the *sfh5Δ* chemogenomic profile and the ability of Sfh5 to bind

heme are considered in aggregate, these properties converge on a hypothesis that Sfh5 functions in some aspect of redox control, and/or regulation of heme homeostasis, under stress conditions induced by exposure to organic oxidants.

## Discussion

Herein, we describe the structural, biophysical and bioinorganic properties of Sfh5 – an atypical Sec14-like protein. We report that Sfh5 is a penta-coordinate high spin $Fe^{3+}$ heme-binding protein with unusual heme-binding and heme Fe-coordination properties. These describe a complex electronic system that is, to our knowledge, unique. As such, Sfh5 is the prototype of a new class of hemoproteins conserved across the fungal kingdom. In contrast to the known priming lipids of other Sec14-like PITPs, heme is not an exchangeable ligand for Sfh5, and the feeble PtdIns-exchange activity of Sfh5 is strongly enhanced in heme-binding mutants – indicating Sfh5 is not designed to function as a bona fide PITP in vivo. While Sfh5 is non-essential for yeast viability under all conditions tested thus far, chemogenomic analyses suggest Sfh5 participates in stress responses to specific (albeit as yet unidentified) environmental organic oxidants. The collective data offer a powerful example of how evolution has used the Sec14-fold as a versatile scaffold for accommodating the binding of a wide variety of ligands, not only lipids, for the purpose of regulating diverse sets of cellular activities.

### Sfh5 is a novel heme-binding protein

Solutions of purified Sfh5 are strikingly reddish-brown in color and this property reflects the fact that Sfh5 contains bound Fe. Soret signatures of purified Sfh5 are consistent with a non-covalently bound heme *b*-Fe complex. The physiological relevance of heme binding is amply demonstrated by the fact that Sfh5 purified from yeast contains non-covalently bound heme, and that purified recombinant Sfh5 orthologs derived from divergent fungal species similarly exhibit Soret profiles diagnostic of bound heme *b*-Fe. Moreover, the accessibility of Sfh5 heme to exogenous ligands such as $CN^-$ and CO indicates a redox-active heme center with an open and available axial coordination site. The heme-bound Sfh5 structure presents a typical Sec14-fold consisting of an N-terminal tripod domain, and a C-terminal domain that forms the 'lipid-binding' pocket of Sfh5 (*Sha et al., 1998*; *Schaaf et al., 2008*). Consistent with the ICP-MS and Soret data, a single non-covalently bound heme *b*-Fe complex is visualized per Sfh5 monomer, and bound heme is sequestered deep within the 'lipid-binding' cavity.

### Structural description of the Sfh5 heme center

A unique feature of the Sfh5 heme center is a complex electronic system where Fe is coordinated by $Y_{175}$ in conjunction with the co-axial $H_{173}$. This system incorporates three aromatic ring systems (the $Y_{175}$ benzyl ring, the $H_{173}$ imidazole, porphyrin) and open-shell iron d-orbitals that indicate extensive delocalization. How this unique structure operates poses an open question. That Fe- and heme-coordinating residues are highly conserved amongst the fungal Sfh5 orthologs reinforces our conclusion that these orthologs represent heme-binding proteins, and predicts that they also share the novel integrated electronic system defined by Sfh5. Moreover, the structural information identifies a heme-binding barcode for this class of Sec14-like proteins.

The absence of a sixth coordinating ligand for the Fe was evident from the structure – confirming Sfh5 heme represents a penta-coordinate system. The vacant site can be occupied by diatomic molecules bound by Sfh5, and is the probable binding site for other potential Sfh5 ligands. The structure also suggests that entry into this vacant coordination site is likely gated by the side chains of two invariant residues of the Sfh5 clade ($K_{188}$ and $K_{192}$) and possibly conformational transitions of a gating helix analogous to that described for other Sec14-like PITPs. Interestingly, only ~4% of known hemoproteins exhibit Tyr as an axial ligand, and the majority of those that are penta-coordinate are catalases (*Li et al., 2011*). However, Sfh5 shows no measurable catalase activity – consistent with the absence of polar side chains near the sixth coordination site capable of stabilizing the oxo-ferryl intermediate of the catalase reaction. These collective results reinforce the structural and biophysical novelty of this site in Sfh5.

## Magnetic properties of the Sfh5 heme center

EPR and Mössbauer analyses demonstrate Sfh5 contains a high-spin S = 5/2 $Fe^{3+}$ heme center, and both the EPR and Mössbauer signatures are typical of high-spin hemes. The electronics of the heme center are responsive to substitution of the co-axial $H_{173}$ with Ala or Tyr, or substitution of $Y_{175}$ with His, in a manner consistent with $H_{173}$ status modulating the $Y_{175}$-Fe interaction. Presumably, this modulation is achieved by stabilization of the Fe-coordinating tyrosinate ion. Mössbauer analyses identify two discrete magnetic states for the Sfh5 heme center, and one state is lost in the Sfh5$^{H173A}$ mutant. One interpretation of the data is that the two states report alternate protonation states for $H_{173}$. However, attempts to demonstrate a pH-dependent shift from one state to the other by adjusting the pH of the protein solution proved unsuccessful – perhaps due to insufficient solvent exposure of this region of the heme coordination motif. Alternatively, the two states might reflect different conformational states of the heme as heme distortions can be associated with stronger axial ligand-heme bonding and changes in electronic structure of the iron center (*Olea et al., 2008*; *Négrerie, 2019*).

## Heme coordination strategies of Sfh5 and hemophores of bacterial pathogens

Hemophores are proteins secreted by bacterial pathogens that function in Fe-acquisition (*Nasser et al., 2016*; *Sheldon and Heinrichs, 2015*). These do so by scavenging heme from the host and, upon receptor-mediated retrieval of the hemophore/heme complex to the bacterial cell surface, releasing that micronutrient to the pathogen. HasA of *Serratia marcescens* and IsDX1 of *Bacillus anthracis* represent prototypical bacterial hemophores (*Wilks and Burkhard, 2007*; *Tong and Guo, 2009*; *Korolnek and Hamza, 2014*; *Philpott and Protchenko, 2016*). While Sfh5 shares no sequence or structural homology to HasA or IsDX1, and Sfh5 is not secreted from the cell, the yeast protein does share interesting parallels in its heme coordination strategy with those hemophores. For example, HasA exhibits tyrosine ($Y_{75}$) as one of two axial ligands, and an adjacent histidine ($H_{83}$) H-bond acceptor stabilizes the tyrosinate ion and increases the affinity of heme binding (*Létoffé et al., 2004*; *Caillet-Saguy et al., 2008*; *Cescau et al., 2007*). Structural analyses of heme-bound HasA in complex with its bacterial surface receptor further suggest that transient protonation of the HasA $H_{83}$ Nδ weakens the H-bond with $Y_{75}$, reduces the affinity of HasA for heme, and facilitates heme release to the receptor (*Krieg et al., 2009*). Other heme-binding NEAT-domain proteins (e.g. IsdA and IsdH-N3 of *S. aureus* and IsdX1 of *B. anthracis*) employ Tyr-Tyr coordination motifs that operate via similar mechanisms (*Grigg et al., 2007*; *Watanabe et al., 2008*; *Ekworomadu et al., 2012*).

## Sfh5 does not exhibit the properties of a heme donor

Yeast cells synthesize heme in the mitochondria, but the mechanisms of intracellular heme transport and compartmentalization remain largely unresolved (*Protchenko et al., 2008*; *White et al., 2013*). In that regard, the interesting parallels between Sfh5 and heme-binding strategies of bacterial hemophores, when considered together with the activities of hemophores in binding and releasing heme on demand, suggested the possibility that Sfh5 functions in intracellular heme distribution. Two lines of evidence do not support this idea, however. First, consistent with the high affinity of Sfh5 for heme, the inability of an avid heme scavenger to withdraw heme from Sfh5 suggests Sfh5 is not designed to donate bound heme to some proteinaceous acceptor. One caveat associated with this interpretation is the formal possibility that Sfh5 donates heme to a highly privileged acceptor cannot yet be excluded. Second, Sfh5 exhibits only low-level PtdIns transfer activity in vitro and is ineffective in stimulating PtdIns 4-OH kinase activity in vivo. Yet, 'heme-less' Sfh5 mutants are resuscitated for both PtdIns-transfer activity in vitro and stimulation of PtdIns 4-OH kinase activity in vivo. These data report a fundamentally antagonistic relationship between heme- and PtdIns-binding where binding of the former is incompatible with exchange of the latter. We speculate that heme-binding locks Sfh5 into what is a substantially 'closed' conformer and that this represents the functional Sfh5 configuration. That holo-Sfh5 crystallizes as a 'closed' conformer is consistent with this view.

## The enigmatic physiological function of the Sfh5 hemoprotein

Sfh5 does not play a major role in yeast heme biology under laboratory conditions – at least not one accompanied by a recognizable phenotype. Yet, this hemoprotein is clearly conserved across the fungal kingdom. What then is the significance of the conserved heme-binding ability of Sfh5? One possibility is that heme represents a structural cofactor required for Sfh5 folding, and that Sfh5 activity does not otherwise leverage iron redox chemistry towards some biological function. The deeply sequestered heme pose in the internal Sfh5 cavity, its non-exchangeability, the fact that purified recombinant Sfh5 is very defective in PtdIns-transfer activity even though only some 1/3 of the molecules are heme-bound, and that Sfh5 is destabilized by removal of bound heme, are all consistent with such a scenario.

Two lines of evidence counter this idea, however. First, as described above, Sfh5 mutants unable to bind heme are well-folded proteins that exhibit robust PtdIns-exchange activity in vitro and are effective in stimulating PtdIns 4-OH kinase activities in vivo. Thus, these mutants are competent to undergo the significant conformational transitions required for PITP activity. Perhaps the corresponding amino acid substitutions exert the dual effects of rendering Sfh5 not only defective in heme-binding, but also sterically permissive for PtdIns-binding. This possibility is strongly supported by our demonstration that wild-type Sfh5 expressed in yeast cells devoid of heme remains incompetent for PITP activity in stimulating PtdIns 4-OH kinase activities in vivo. An innate steric incompetence of Sfh5 for PtdIns-binding provides yet another indication the protein is not designed to function as a PITP.

Second, chemogenomic analyses identify a specific 'heme-requiring' signature for Sfh5-deficient yeast challenged with the organic oxidant oltipraz – an organosulfur compound that is not only used to treat schistosomiasis, but also exerts anti-cancer effects by inducing phase II carcinogen detoxification systems (*Velayutham et al., 2007*; *Abdul-Ghani et al., 2009*). Those data argue that heme binding by Sfh5 is physiologically relevant, but only when yeast are exposed to specific oxidative stressors. In that regard, the human pathogen *Cryptococcus neoformans* offers potential insight. This fungus causes lung infections that frequently progress to cryptococcal encephalomeningitis in immune-suppressed individuals and represents a leading cause of death in AIDS patients (*Charlier et al., 2008*). *C. neoformans* expresses three Sec14-like proteins – a restricted cohort as fungi typically express at least five. Two of the *C. neoformans* Sec14-like proteins are orthologs of the canonical PtdIns/PtdCho-exchange PITP Sec14, and one is an Sfh5 ortholog (*Chayakulkeeree et al., 2011*). Perhaps Sfh5 hemoproteins participate in stress responses to organic oxidants encountered by commensal and pathogenic fungi in host contexts, and by non-pathogenic yeast such as *Saccharomyces* in the wild.

## Versatility of the Sec14-fold

Whereas the biological activity of Sfh5 remains enigmatic, what these collective results powerfully highlight is how evolution uses the Sec14-fold as a versatile scaffold for accommodating the binding of a wide variety of ligands. Our discovery of a novel class of Sec14-like hemoproteins now extends the diversity of ligand binding by members of the Sec14 PITP-like superfamily to molecules that are not lipids. That Sfh5 and its fungal orthologs exhibit an unusual heme coordination mechanism that includes a planar porphyrin, a redox-active iron, and co-axial Tyr and His residues packaged as an integrated electronic system, emphasizes the high complexity of ligand coordination possible with the Sec14-fold. Moreover, that this unique electronic system offers an available axial binding site suggests Sfh5 has a capacity to bind as yet unidentified substrate(s) and to execute new heme-based redox chemistry. It is an exciting proposition that the Sfh5-fold might ultimately prove amenable to the rational engineering of new redox nanoreactors designed for targeted modification of interesting organic molecules.

# Materials and methods

## Yeast strains, plasmids, and media

The yeast strains used in this study were CTY182 (*MATa ura3-52 lys2-801 Δhis3-200*), CTY1-1A (*MATa ura3-52 lys2-801 Δhis3-200 sec14-1^ts*), CTY558 (*MATα ade2 ade3 leu2 Δhis3-200 ura3-52 sec14Δ1::HIS3/pCTY11*) (*Bankaitis et al., 1989*; *Cleves et al., 1991*). Yeast strains constructed in

this study included: YDK100 (CTY182 *sfh5Δ::KanmX4*), VBY64 (CTY182 *sfh5Δ::KanmX4* [His6]*SFH5:: HIS3*), YDK112 (CTY182 *P_{PMA1}::SFH5KanMX4-PMA1*), YDK113 (*MATa ura3-52 lys2-801 Δhis3-200 hem1Δ::NatMX6*) and YDK 114 (*MATa ura3-52 lys2-801 Δhis3-200 sec14-1^{ts} hem1Δ::NatMX6*). The *hem1Δ(6D)* mutant was a gift from Caroline Philpott (NIH). Structural genes encoding Sfh5 homologs were PCR-amplified from *S. cerevisiae*, *C. albicans* and *C. glabrata* genomic DNA and subcloned into pET28b+ vectors as *Nco*I and *Sac*I restriction fragments such that an 8 × histidine tag was incorporated at the N′ terminus of each open reading frame. The Q5 site-directed mutagenesis kit was used in all site-directed mutagenesis experiments (New England Biolabs). The *HRG-1* coding region was amplified from cDNA prepared from *C. elegans* mRNA and subcloned into a modified pYES-DEST52 vector as a SacII/SphI fragment thereby placing *HRG-1* expression under *GAL1* promoter control. As control, *SFH5* from *S. cerevisiae* was similarly subcloned in parallel. The identities of all constructions were confirmed by DNA sequencing (Eton). Genetic methods and media used were previously described (*Kearns et al., 1998*; *Khan et al., 2016*).

## Protein expression and purification

Sec14p was purified as described (*Schaaf et al., 2008*). For Sfh5p, a pET28b(+) vector harboring 6 × His -Sfh5 was transformed into *E. coli* BL21 (DE3) cells and grown in Luria broth (LB) at 37°C. Upon the culture reaching $OD_{600} = 0.5$, protein expression was induced with isopropyl β-D-1-thiogalactopyranoside (80 µM final concentration), and the culture was incubated overnight at 16°C with shaking. Cells were harvested by centrifugation at 4°C, cell pellets were resuspended in buffer A (300 mM NaCl and 25 mM $Na_2HPO_4$, pH 7.0) with 2 mM PMSF and passed twice through a French Press at 10,000 psi. The crude lysate was clarified by centrifugation at 2800 g and then at 27,000 g. The clarified fraction was incubated with Co-NTA resin (TALON, Clontech) in the presence of 10 mM imidazole followed by wash with buffer A plus 20 mM imidazole. Sfh5 was eluted using a four-step gradient of buffer A supplemented with 50 mM, 100 mM, 200 mM and 400 mM imidazole. Fractions enriched in Sfh5 were reddish brown, and protein purity was monitored by SDS-PAGE. Peak fractions were pooled and subjected to three rounds of dialysis against buffer A, followed by size-exclusion chromatography on Sephadex 200 column (GE Healthcare). Sfh5 was stored at −80°C.

For isotopic labeling of Sfh5 with [57]Fe-labeled heme, *E. coli* BL21(DE3) cells were transformed with the $His_6$-Sfh5-pET28b(+) plasmid and a pre-inoculum was grown overnight in LB medium at 37°C. Cells were washed twice in modified M9 minimal medium (42 mM $Na_2HPO_4$, 22 mM $KH_2PO_4$, 8.5 mM NaCl, 18.5 mM $NH_4Cl$, 2 mM $MgSO_4$, 0.1 mM $CaCl_2$ with 20 mM glucose) and grown in the same medium supplemented with 40 µM [57]$FeCl_3$ at 37°C until the culture reached an $OD_{600} = 0.5$. Sfh5 expression was induced with IPTG (final concentration 100 µM), and cells were incubated for 36 hr at 16°C with shaking. Cells were harvested and Sfh5 purified as described above. A 6 × His tagged version of Sfh5 was expressed in yeast under the control of the constitutive *PMA*1 promoter. Yeast cells were lysed by three rounds of passage through an LM20 microfluidizer (Siemens) at 30,000 psi. Sfh5 purification followed essentially the same procedure as that described for recombinant protein purification from *E. coli*.

## Analysis of Sfh5-bound metal

The metal contents of Sfh5 proteins were determined by ICP-MS. Proteins were digested overnight at 80°C in the presence of trace-metal grade nitric acid. Precipitated material was removed by centrifugation and the supernatant collected for analysis. Perkin Elmer Elan DRC II ICP-MS machine was used in collision cell mode (He, 4.5 mL/min) with a skimmer to minimize polyatomic inferences. A standard Micromist nebulizer (Glass Expansion, Australia) was used to introduce the sample. The dwell time for [56]Fe analysis was 100 ms.

## Mass spectrometry

Proteins were desalted with a C4 ZipTip (MilliporeSigma) and so prepared to achieve concentrations that ranged from 2 to 10 mg/ml. For matrix-assisted laser desorption ionization time of flight (MALDI TOF) mass spectrometry, a matrix solution composed of 10 mg/ml α-cyano-4-hydroxycinnamic acid (CHCA) in 50% acetonitrile and 0.1% trifluoroacetic acid (v/v) was made. One µL of the CHCA matrix was mixed with 1 µL of protein and allowed to dry at 30°C for 15 min. Approximately 1000 laser

shots were collected for each sample on a Bruker Ultraflextreme MALDI TOF/TOF mass spectrometer with positive ion setting in the reflector mode. Hemin chloride (Sigma) was used as standard.

## Sfh5 crystallization

Diffraction quality crystals were grown under conditions where 300 nL of a 10 mg/ml Sfh5 preparation was mixed with an equal volume of 0.1 M potassium thiocyanate, 0.1 M sodium citrate (pH 4.2) and 15.5–26% PEG 4000. Prior to data collection, crystals were cryoprotected with 20% (v/v) ethylene glycol, and flash-frozen in liquid nitrogen for data collection.

## Model building and refinement

Data collection was performed on the 23-ID beamline of Advance Photo Source (APS) at Argonne National Laboratory, Chicago, IL, USA, with three different crystals at 8 keV (1.5498 Å). Data were indexed, integrated, merged and scaled with anomalous flags using *HKL2000* (*Otwinowski and Minor, 1997*). *Xtriage* predicted the anomalous signal for iron from the merged dataset to extend to 3.6 Å. For initial phasing, the resolution was set at 3.2 Å. The Sfh5 structure was solved using Phenix Autosol Wizard by single wavelength anomalous diffraction method using iron anomalous signal from three heavy atom positions in the asymmetric unit (*Adams et al., 2010*). The data were submitted to Autobuild and an initial model was built. For subsequent refinement and model building, we chose a minimal complete subset of data with the best scaling statistics and resolution cut at 2.9 Å based on the intensity and CC1/2 analyses (data statistics for both phasing and refinement sets are given in *Table 1*). A well-defined electron density consistent with heme *b* was identified which correlated with the heavy atom positions. The coordinates and geometry definition files for the heme ligands were generated with Phenix Elbow (*Moriarty et al., 2009*) without any artificial restraints. The initial model was improved by manual rebuilding in COOT (*Emsley and Cowtan, 2004*), and the final model was obtained after iterative cycles of model building and Phenix refinement. Statistics related to data collection and refinement are given in *Table 1*. The final structure was validated, and the atomic coordinates and structure factors for the structure were deposited in the Protein Data Bank with the accession code 6W32.

## Protein preparation for in silico analysis

Protein models were prepared using the *Protein Preparation Wizard* panel in the Schrödinger suite (2018–1, Schrodinger, LLC, Mew York, NY, 2018). The Sfh5 structure was optimized with the OPLS_2005 forcefield in the Schrödinger suite to relieve all atom and bond strains found after adding all missing side chains and/or atoms. The Sfh5::PtdIns model was generated by structural overlay of Sfh1::PtdIns complex (PDB ID 3B7N; *Schaaf et al., 2008*) on Sfh5 monomer. The heme group of Sfh5 was replaced by PtdIns, which was extracted from Sfh1. The Sfh5::PtdIns complex was energy minimized to relieve Sfh5 and PtdIns atoms of any van der Waal steric clashes and complex was optimized for electrostatic interactions. Molecular graphics and analyses were performed with UCSF Chimera and Schrödinger's Maestro program.

## Hydrodynamic analyses

Equilibrium sedimentation of Sfh5 in 50 mM $NaPO_4$, 300 mM NaCl (pH 7.0) was performed at 4°C on an Optima XL-A Analytical Ultracentrifuge (Beckman Coulter). Equilibrium radial distributions at 10,000 and 15,000 RPM of Sfh5 at concentrations of 10, 5 and 2.5 µM were recorded via UV absorbance at 280 nm. All data (two speeds, three concentrations) were fit globally to a single ideal species with floating molecular weight using nonlinear least squares using the HETEROANALYSIS software 1. Protein partial specific volume and buffer density were calculated with SEDNTERP 2.

## UV-visible absorbance spectroscopy

Absorbance measurements were performed on a Cary Varian Bio 100 UV-vis spectrophotometer (Varian) at room temperature. The slit width for all measurements was 1 nm and spectra were collected at a rate of 500 nm per minute. Quartz cuvettes were used for all measurements.

## Pyridine hemochromagen assay

Heme was identified and quantified using a pyridine hemochromagen assay performed as described (*Barr and Guo, 2015*). Briefly, 0.5 ml of protein solution of fixed concentration (30 µM) was oxidized by adding a 0.5 ml cocktail of 0.2 M sodium hydroxide, 40% (v/v) pyridine and 500 µM potassium ferricyanide, and a UV-vis absorption spectrum was collected. Following addition of 1 mg of sodium dithionite, a second spectrum was collected. Appearance of α and β bands at 557 and 525 nm respectively, in the $Fe^{2+}$ state were taken as evidence of heme *b*. Using known extinction coefficients for pyridine hemochromagens, heme contents were quantified for each sample.

## EPR spectroscopy

Purified proteins were collected in Wilmad Suprasil EPR tubes and gently frozen in liquid $N_2$ and stored until data acquisition. EPR spectra involved a 5000G sweep on an X-band EMX spectrometer (Bruker Biospin Corp) with a bimodal resonator and cryostat for maintaining a low temperature of 4K. Average microwave frequency, 9.38 GHz; microwave power, 0.2 W; modulation amplitude, 10 G; average time = 300 s. All spectra were normalized and plotted using SpinCount software (http://www.chem.cmu.edu/groups/heindrich). A 1 mM solution of $CuSO_4$-EDTA was used as a standard.

## Mössbauer spectroscopy

Low-field (0.5 T), low-temperature (5K) Mössbauer spectra were collected using a model MS4 WRC spectrometer (See Co. Edina, MN). Variable field (0–6 T), 4.2 K spectra were collected using a model LHe6T spectrometer. Both instruments were calibrated using an α-Fe foil at room temperature. Approximately 800 µl of recombinant proteins (concentration = 1.5 mM) were collected in a Mössbauer cup, frozen over liquid nitrogen and stored at −80°C until use. Spectra were simulated using WMOSS (http://www.wmoss.org).

## Peroxidase assays

Peroxidase assays were performed as described by *Maehly and Chance, 1954*. Briefly, the appropriate protein samples were incubated with pyrogallol and its conversion to purpugallin was monitored spectrophotometrically at 420 nM wavelength as a function of time. One unit of peroxidase activity was defined as formation of 1 mg of purpurogallin in 20 s from the substrate at pH 7.0 at room temperature. Activities were normalized to protein heme content as determined by ICP-MS and pyridine hemochromagen assay.

## PtdIns transfer assays

The assays were performed using rat liver microsomes as [$^3$H]-PtdIns donor and PtdCho liposomes as acceptor as previously described (*Schaaf et al., 2008*). All incubations were for 30 min at 37°C.

## In vitro heme transfer

Apo-myoglobin was prepared by methy-ethyl ketone extraction of heme from equine skeletal myoglobin as previously described (*Di Iorio, 1981*). Heme-free apo-myoglobin (10 µM) was incubated with hemin (30 µM) and Sfh5 (35 µM) as indicated for different time periods and resolved on non-reducing SDS-PAGE gels run in duplicate. One gel was stained with Coomassie Blue and other was transferred onto a nitrocellulose membrane and probed for peroxidase activity using SuperSignal West Femto Maximum Sensitivity Substrate reagent (Thermo Fisher). Heme transfer from Sfh5 donor to apo-myoglobin acceptor was monitored by peroxidase activity in the myoglobin bands (*Feissner et al., 2003*).

## Solvent accessible surface area calculations

SASA of Sfh5 with and without bound heme were calculated using the Schrodinger binding SASA script. This routine calculates solvent accessible surface area of ligand and receptor before and after receptor-ligand binding. A default cutoff of a 5 Å radius around the heme ring system was used to compute SASA of the heme binding site before and after heme binding to Sfh5.

## Acknowledgements

This work was supported by grants R35 GM131804 from the National Institutes of Health (NIH) and BE-0017 from the Robert A Welch Foundation to VAB. The Laboratory for Molecular Simulation and High Performance Research Computing at Texas A and M University provided software, support, and computer time. JW and PAL were supported by grants GM127021 from the NIH and A1170 from the Robert A Welch Foundation to PAL, JPW and AL were supported by NIH grant DP2GM123486 awarded to AL, DME was supported by NIH grant P50 GM082545, and GG, AA, IK and JS were supported by Welch Foundation grant A-0015 awarded to JS.

We are grateful to the reviewers for their constructive criticisms and for suggesting the *hem1Δ*/ ergosterol experiment to test whether Sfh5 PITP activity is resuscitated in yeast cells devoid of heme. We also thank Tatyana Igumenova (Dept. Biochemistry and Biophysics, TAMU) for helpful discussions, and the staff at GM/CA beamline 23ID-D of the Advanced Photon Source (Argonne National Laboratory) for assistance during the X-ray data collection. GM/CA@APS has been funded in whole or in part with Federal funds from the National Cancer Institute (ACB-12002) and the National Institute of General Medical Sciences (AGM-12006). This research used resources of the Advanced Photon Source, a U.S. Department of Energy (DOE) Office of Science User Facility operated for the DOE Office of Science by Argonne National Laboratory under Contract No. DE-AC02-06CH11357. The authors declare no conflicts of interest.

VAB dedicates this paper to the memory of his friend and colleague, the incomparable Michael Wakelam, whose quiet quality elevated the scientific standard in our lipid signaling community.

## Additional information

### Funding

| Funder | Grant reference number | Author |
| --- | --- | --- |
| National Institute of General Medical Sciences | R35 GM131804 | Vytas A Bankaitis |
| Welch Foundation | BE-0017 | Vytas A Bankaitis |

The funders had no role in study design, data collection and interpretation, or the decision to submit the work for publication.

### Author contributions

Danish Khan, Conceptualization, Data curation, Formal analysis, Validation, Investigation, Visualization, Methodology, Writing - original draft, Writing - review and editing; Dongju Lee, Formal analysis, Investigation, Methodology, Writing - review and editing; Gulcin Gulten, Debra M Eckert, Conceptualization, Data curation, Formal analysis, Investigation, Methodology, Writing - original draft, Writing - review and editing; Anup Aggarwal, Data curation, Formal analysis, Validation, Investigation, Visualization, Methodology, Writing - original draft, Writing - review and editing; Joshua Wofford, John W Patrick, Formal analysis, Investigation, Methodology, Writing - original draft; Inna Krieger, Conceptualization, Data curation, Formal analysis, Investigation, Methodology, Writing - review and editing; Ashutosh Tripathi, Conceptualization, Formal analysis, Investigation, Visualization, Methodology, Writing - original draft, Writing - review and editing; Arthur Laganowsky, Data curation, Formal analysis, Funding acquisition, Validation, Investigation, Methodology, Writing - original draft, Project administration, Writing - review and editing; James Sacchettini, Conceptualization, Data curation, Formal analysis, Funding acquisition, Validation, Investigation, Visualization, Methodology, Writing - original draft, Project administration, Writing - review and editing; Paul Lindahl, Conceptualization, Data curation, Formal analysis, Supervision, Funding acquisition, Validation, Investigation, Visualization, Methodology, Writing - original draft, Project administration, Writing - review and editing; Vytas A Bankaitis, Conceptualization, Resources, Formal analysis, Supervision, Funding acquisition, Investigation, Methodology, Writing - original draft, Project administration, Writing - review and editing

Author ORCIDs

Danish Khan (iD) http://orcid.org/0000-0002-0650-3990
James Sacchettini (iD) http://orcid.org/0000-0001-5767-2367
Vytas A Bankaitis (iD) https://orcid.org/0000-0002-1654-6759

Decision letter and Author response
Decision letter https://doi.org/10.7554/eLife.57081.sa1
Author response https://doi.org/10.7554/eLife.57081.sa2

---

## Additional files

### Supplementary files
- Supplementary file 1. Biochemical and biophysical properties of Sfh5 and mutant derivatives.
- Transparent reporting form

### Data availability

Diffraction data have been deposited in PDB under the accession code 6W32. All data generated or analysed during this study are included in the manuscript and supporting files.

The following dataset was generated:

| Author(s) | Year | Dataset title | Dataset URL | Database and Identifier |
|---|---|---|---|---|
| Sacchettini J, Gulten G, Aggarwal A, Krieger I, Danish K, Bankaitis AV | 2020 | Crystal structure of Sfh5 | https://www.rcsb.org/structure/6W32 | RCSB Protein Data Bank, 6W32 |

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
