## [Decision Letter]

**Acceptance summary:**

This manuscript shows how evolution has reshaped a binding pocket that normally harbours phospholipid, into a heme-binding pocket. It will be very interesting to investigate the physiological role of heme binding in the future; the crystal structure and thorough biochemical characterization of the Sfh5 protein show an unambiguous and unexpected twist in the evolution of lipid binding proteins that is of high interest to a wide readership.

**Decision letter after peer review:**

Thank you for submitting your article "A Sec14-like phosphatidylinositol transfer protein paralog defines a novel class of heme-binding proteins" for consideration by *eLife*. Your article has been reviewed by four peer reviewers, including Benoit Kornmann as the Reviewing Editor and Reviewer #1, and the evaluation has been overseen by a Reviewing Editor and Cynthia Wolberger as the Senior Editor. The following individuals involved in review of your submission have agreed to reveal their identity: Amit R Reddi (Reviewer #2).

The reviewers have discussed the reviews with one another and the Reviewing Editor has drafted this decision to help you prepare a revised submission.

Summary:

This manuscript reports the unusual reconfiguration of a phospholipid binding pocket into a heme binding pocket. Structural analysis of the PhosphatidylInositol Transfer Protein (PITP) Sfh5 shows that the protein binds heme with an unusual electronic arrangement in anpocket that is normally occupied by PI. Analysis of the residues involved in binding and of the affinity to heme indicate that heme is redox active, could adsorb another ligand on one side of the iron atom, and is bound to the protein almost irreversibly.

Examining published pharmacogenomics screens unveils a link between Sfh5 and a potential redox response.

Notwithstanding some problems in crystallography interpretation (see below), all four reviewers found that the study was carefully executed, and that the repurposing of a lipid-binding pocket to a heme-binding pocket was surprising and interesting. They felt however that the biological function of heme-binding by Sfh5 was insufficiently investigated. In particular, reviewers 1, 2 and 3 felt that it was at this stage premature to conclude that Sfh5 had lost its ability to bind PI or that heme-binding was not regulating PI binding. A role of Sfh5 as a PITP should therefore not be discounted at this point given how easy it is to resuscitate this function, and given that the concentrations of heme and PI might allow a fair fraction of Sfh5 to be PI-bound in vivo.

Essential revisions:

Reviewer 2 proposes an experiment that could shed light on a possible role of Sfh5 as heme-regulated PITP. Heme levels can be titrated up and down in cells. To do this, heme is first made dispensable by providing ergosterol in the medium (the essential function of heme is in ergosterol biosynthesis). In this condition the aminolevulinate synthase HEM1 can be deleted. Heme levels can then be tuned by adding back given amounts of aminolevulinic acid (which is imported into cells).

Since you show that a heme-binding deficient Sfh5 mutant recovers its PITP activity (in Sec14 delete cells), it is to be expected that wildtype Sfh5 will function as PITP in heme-less cells. Here, by increasing the amount of aminolevulinate (and thus of intracellular heme), Sfh5 should gradually lose its PITP activity. Such an experiment should be quite straightforward and could shed light on the levels of heme necessary to inhibit the PITP function of Sfh5.

Moreover, reviewer 4 found substantial problems with the validation of the structure based on the density map (see their point 4). The map has also been checked by an independent expert, who saw nothing wrong with the number of molecules per unit cell, but agreed that there are a lot of issues with the structure that need to be fixed.

- The validation report also flagged issues with the heme geometry that should be fixed.

- The large number of clashes and protein geometry issues come from poorly traced chains in the C-terminal residues of chains A and C, and in all three copies in the vicinity of residues 40-45. One can see very clearly from the density that the fit is incorrect and violates various geometric constraints. These are all fixable and probably don't affect the interpretation.

Reviewer #1:

This manuscript describes the recombinant expression, purification and structural analysis of Sfh5, a Sec Fourteen Homolog, and the surprising discovery that, unlike Sec14 and other related proteins, Sfh5 doesn't appear to bind the phospholipid phosphatidylinositol (PI), but instead binds heme tightly and through an unusual network of bonds. The physiological implication of this binding is not yet understood, but experimental indications allow the authors to speculate on possible roles. First, because the iron center of heme is pentavalent, and that a cavity exists where the sixth bond would be, suggesting that a secondary substrate could bind here (like O2 in hemoglobin). Second, the analysis of a chenogenomics screen shows that Sfh5 mutant cells are sensitive to heme-depleting drugs.

This is a very intriguing story that certainly asks more questions than it answers. It is carefully executed and exhaustive. It will certainly lay the ground for further research.

Essential revisions:

- The authors speculate that Sfh5 doesn't bind PI in a physiological setting and is unlikely to exchange heme for PI, given the difference in affinity to both metabolites. This is somewhat contradictiry with the fact that PI binding can apparently be rescued by abrogating heme binding, indicating that PI binding has been kept throughout evolution. The authors mention a "barcode" of aminoacids necessary for PI binding, which all have been kept in Sfh5. Moreover, mutating one amino acid of this barcode allows a heme-binding-deficient mutant to regain heme binding to some extent, suggesting that it is PI binding that prevents heme binding in the heme-defective mutant. So what are the amino-acids of this barcode? Are they conserved, and if so, why?

If the "resuscitable" PI-binding ability of Sfh5 is conserved, then it probably means that PI-binding is an important property of the protein.

Also, the concentrations of heme and PI might be very different wherever Sfh5 is located, which might compensate for the different inherent affinities. Finally, the authors rightly acknowledge that "the formal possibility that Sfh5 donates heme to a highly privileged, and as-yet-unidentified, acceptor cannot be excluded", they do not acknowledge the possibility that Sfh5 binding to a third party factor (not necessarily a heme-acceptor) might decrease its affinity for heme in vivo. These points suggest that it could be premature to dismiss the possibility that heme/PI exchange might actually happen in vivo. This possibility does not alter the interest of the present study and can be addressed in further studies.

- This manuscript talks to scientists with very different sets of expertise. The thorough characterization of the biophysical properties of Sfh5 could in general do with more complete explanations of the various methods in use. For instance:

Soret spectra are calibrated against mock-expressing bacterial lysate. This information could go in the text or figure legend rather than the Materials and methods section.

"The isostructural A10/T4 gating helices of Sfh1 and Sfh5 were positioned similarly across the opening to the internal cavity". The corresponding figure could benefit from better description (no labels on the helices).

"tyrosine coordination was consistent with the results of safranin T dye-reduction assays which estimated an unusually low thermodynamic reduction potential for Sfh5 heme (-330 ± 3 mV vs. NHE)." This could require better explanations (e.g. NHE is not described anywhere).

Reviewer #2:

The authors describe an unusual but very interesting member of a class of Sec14-like phosphatidylinositol transfer proteins (PITP), Sfh5. They find that Sfh5 is a novel heme binding protein and, in fact, does not function in activities typically associated with PITPs. Most interestingly, they find that variants of Sfh5 that cannot bind heme restores PITP activity. The major focus of the study is on the biophysical characterization of Sfh5 and its interactions with heme. The work is technically sound, beautifully integrating rigorous structural, spectroscopic, and biochemical characterization of Sfh5, and the manuscript is very well written. However, the studies may not have broad appeal due to the lack of cell biological data demonstrating the physiological significance of heme-Sfh5 interactions.

Essential revisions:

1) The paper is technically sound but may be of limited impact due to the lack of data establishing a physiological function for heme binding. In this sense, significant efforts must be made to establish a role for heme binding. As it currently stands, the work may be more appropriate for a specialized audience of biochemists.

2) The authors should comment on how the PITP activity of the heme-binding mutants of Sfh5 compare to other Sec14-like PITPs? These mutants may have more PITP activity than WT Sfh5, but is the activity comparable to other more typical Sec14-type PITPs?

The following suggestions are outside the scope of the current study as presented. I hope the comments may prove useful to the authors as they continue their studies on this interesting protein.

3) Do the authors think heme plays a regulatory role in Sfh5 phosphatidylinositol transfer activity? For instance, it is mentioned that only 1/3 of the protein is heme loaded. Does increasing heme synthesis decrease PITP activity in vivo? If the authors fed WT cells porphobilinogen (PBG), which would increase heme levels, or titrate aminolevulinic acid (ALA) in *hem1∆* cells, is there a dose dependent decrease in Sfh5-related PITP activity in vivo? An experiment such as this demonstrating in vivo regulation of PITP activity by heme would greatly elevate the significance/impact of the current work.

Note, in *Saccharomyces cerevisiae*, ALA synthase is not the rate-limiting enzyme for heme synthesis, as it is in other eukaryotes. Thus, to drive heme synthesis in WT cells, one must use PBG.

4) The authors make the intriguing comment that there may be a privileged acceptor that Sfh5 may donate heme to. Related to this point, I am wondering if heme reduction could induce its dissociation? I understand that the myoglobin transfer experiments were done with ferric heme. Could heme dissociate and transfer to myoglobin if it were reduced with dithionite?

Sfh1 in some sense reminds me of yeast Dap1, a homolog of mammalian PGRMC1 and PGRMC2. Dap1 is a low potential hemoprotein (~-300 mV) that binds heme with a tyrosine ligand. It was thought that Dap1 might be a heme chaperone for ER localized Cytochrome P450s, although this is controversial. *dap1∆* exhibits fluconazole and MMS sensitivity, in part due to defects in ergosterol biosynthesis, which requires a P450 enzyme. The authors may want to consider looking at fluconazole sensitivity, or defects in the ergosterol biosynthetic pathway. There was recently a paper in Nature describing role of PGRMCs in nuclear heme trafficking in adipocytes.

Reviewer #3:

This study demonstrates that a member of the large Sec14-like phosphatidylinositol transfer protein (PITP) family, Sfh5, binds heme. This is the first member of this family found to bind heme. The structure of Sfh5 bound to heme is presented and the interactions of the protein with heme and Fe chemistry of the bound heme are carefully examined. While it is fascinating (and surprising) that a PITP binds heme, the study does not shed much light on the functional significance of heme binding by Sfh5. It argues that the protein is unlikely to be a phosphatidylinositol-heme exchanger and shows that Sfh5 does not efficiently deliver heme to the heme-acceptor apo-myoglobin, consistent with the idea that Sfh5 does not supply heme to other proteins. Instead, it suggests that heme bound Sfh5 somehow participates in a cellular response to oxidants but offers no insight into how this might occur. Even without this, I think the finding that PITPs can bind heme is an important discovery and this study could be appropriate for *eLife*. I have two major concerns.

1) The affinity of Sfh5 for heme should be examined in more detail. This study suggests bound heme is not a readily exchangeable, but this is difficult to square with the statement in the discussion that only 1/3 of Sfh5s are heme-bound. The fraction of endogenous Sfh5 bound to heme should be determined and shown. It is important that these measurements be made with protein lacking a His-tag since this tag bind heme. If it is correct that bound heme is not readily exchangeable but only a fraction of the protein is heme-bound, some effort should be made to explain this. Is the on rate very slow?

2) The study demonstrates that heme bound by Sfh5 is accessible to small ligands like CN^-^, indicating that the heme is redox active, but does not explain how changes in the redox state of bound heme could be physiologically relevant. How could these changes be sensed by other proteins? Or how do the authors think bound heme could significantly contribute to cytoplasmic redox chemistry?

Reviewer #4:

Khan et al, report a novel heme-binding function for a member (Sfh5) of a class of yeast proteins (PITPs) that normally transfer lipids. They report a crystal structure of heme-bound Sfh5 and thoroughly describe the biophysical and structural properties of the heme-bound complex using absorption, EPR, and Mössbauer spectroscopies. Data presented indicate that wild-type Sfh5 neither exchanges phosphatidylinositol nor heme. Experiments indicate that yeast deficient of Sfh5 show no significant functional phenotype. However, mining of a "comprehensive HIPHOP chemogenomic study" lead the authors to propose that Sfh5 may be involved in controlling "some aspect of redox control, and/or regulation of heme homeostasis, under stress conditions induced by exposure to organic oxidants" – a hypothesis that was not experimentally explored in this study.

The strengths of this manuscript are the novelty that a Sec14-like protein can bind heme, and the thorough characterization of the structural and biophysical basis for heme binding. The significance of the "unusual heme-binding arrangement" described in detail in this study is not readily apparent. Furthermore, no biological function was detailed in this study that is supported by data. A more robust functional role for Sfh5's heme-binding properties would strengthen this story.

Essential revisions:

1) Introduction: The authors provide a lot of information about the lipid-dependent activities of the PITPs, given that their protein of interest (Sfh5) does not bind lipid and does not appear to function in the manner consistent with PITP function. Since it appears that the function of Sfh5 is unknown (which is interesting), perhaps this should be stated up front along with any information known about its biology in yeast and other eukaryotic homologs-then more briefly mention the functions of related Sec14-like PITPs and lipid-dependent activities as a possible function (i.e., present a hypothesis). The first paragraph of the Results section could be moved up to the Introduction.

2) Most of the work to show that Sfh5 binds heme was performed using protein expressed in bacteria, which produced heme-bound Sfh5 that copurifed during purification steps. One experiment is shown to link heme binding to Sfh5 in yeast cells (Figure 1C, lower right panel) but the data signal-to-noise is very low for the yeast protein compared to the bacterial protein and no control is provided to show the heme interaction in the yeast pull-down experiment is specific. Can growth conditions for bacterial or yeast grown be discovered that allow purification of heme-free Sfh5-or biochemical stripping/unfolding/repurification of heme-free protein-which can then be used in heme binding analysis via absorbance or ITC measurements and phosphatidylinositol binding assays?

3) There is no functional evidence presented to prove that Sfh5, which is expressed at low levels and shows no deletion functional phenotype, or its unique heme-binding properties are important in yeast or any other organism. Is it possible Sfh5 could be vestigial in yeast? Do related proteins from other organisms (i.e., those that evolutionarily precede yeast) express Sfh5 with higher expression and more robust heme-dependent functional activities. If Figure 8—figure supplement 2 provides the only insight into the potential functional relevance of Sfh5, the authors should consider moving it to a main figure.

4) Crystallography: The authors show that Sfh5 is a homodimer using AUC; however, no dimerization was observed in the crystal structure. An issue related to this that should be addressed: when the sfh5_map_coeffs.mtz file provided by the authors is analyzed in CCP4, the Matthews_coef analysis (estimate of the number of asymmetric units in the unit cell) indicates there should be 4 molecules (e.g., two dimers?) in one unit cell with a solvent content ~52.2%. However, the authors solved the structure with 3 molecules in one unit cell. Other issues that should be addressed pertaining to the structure refinement include (1) the Ramachandran outliers in the validation report is 0.5%, which is inconsistent with 0% reported by the authors in the Table 1; (2) the percent of residues with favored Ramachandran is 86%, which can likely be improved; and (3) the R-factor gap (R_work_/R_free_) is large. Finally, the number of atoms in the protein (129) and ligand (5) reported by the authors in Table 1 needs to be corrected.

5) Figure 7: The authors prepared apo-myoglobin by methy-ethyl ketone extraction of heme from equine skeletal myoglobin for use in heme-transfer assays to determine if Sfh5 can exchange heme via detection of heme transfer to apo-myoglobin. However, they do not show a critical positive control showing that free heme can bind to their prepared apo-myoglobin to demonstrate the protein is functionally capable of binding heme.

---

## [Author Response]

Essential revisions:Reviewer 2 proposes an experiment that could shed light on a possible role of Sfh5 as heme-regulated PITP. Heme levels can be titrated up and down in cells. To do this, heme is first made dispensable by providing ergosterol in the medium (the essential function of heme is in ergosterol biosynthesis). In this conditions the aminolevulinate synthase HEM1 can be deleted. Heme levels can then be tuned by adding back given amounts of aminolevulinic acid (which is imported into cells).Since you show that a heme-binding deficient Sfh5 mutant recovers its PITP activity (in sec14 delete cells), it is to be expected that wildtype Sfh5 will function as PITP in heme-less cells. Here, by increasing the amount of aminolevulinate (and thus of intracellular heme), Sfh5 should gradually lose its PITP activity. Such an experiment should be quite straightforward and could shed light on the levels of heme necessary to inhibit the PITP function of Sfh5.

Please see the response to reviewer 2, point 3.

Moreover, reviewer 4 found substantial problems with the validation of the structure based on the density map (see their point 4). The map has also been checked by an independent expert, who saw nothing wrong with the number of molecules per unit cell, but agreed that there are a lot of issues with the structure that need to be fixed.

The geometry restraints file used for the heme was based on ideal geometry, and refinement was run in Phenix with default data/geometric weights. The refined heme fits the density very well but has slight deviation of the bond lengths RMSZ = 2.02-2.11 for ideal lengths (greatest deviation is 0.16 Å) and greatest bond angle deviation from ideal is by 5.99°. It is not uncommon for heme to have slightly distorted geometries when bound to proteins. (For examples in PNAS May 6, 2014 111 (18) 6570-6575; JBC June 18, 2004279, 26489-26499.)

-The validation report also flagged issues with the heme geometry that should be fixed.-The large number of clashes and protein geometry issues come from poorly traced chains in the C-terminal residues of chains A and C, and in all three copies in the vicinity of residues 40-45. One can see very clearly from the density that the fit is incorrect and violates various geometric constraints. These are all fixable and probably don't affect the interpretation.

The editor is correct to point out that chains A andC have problems, and we should have explained this in the manuscript (and now we do). The two chains are highly disordered in the crystal lattice (except for core of the protein where the heme is bound). This was observed for several diffraction data sets including the SAD. However, chain B is very well ordered except for residues 40-45 and we used Chain B for our interpretations in the manuscript. Chain B has 100 % model fit to the data according to RSRZ validation score, no Ramachandran outliers, and only Tyr34 ring clashes with Leu205 and Leu33 side chains hydrogens. Majority of listed clashes are with hydrogens (added to the model by a program for validation), which cannot be accounted for at this resolution, and are not present in the deposited model. Also, majority of the clashes are in regions with poor density, and where there are sidechains that could exist in multiple conformations.

Reviewer #1:[…]Essential revisions:- The authors speculate that Sfh5 doesn't bind PI in a physiological setting and is unlikely to exchange heme for PI, given the difference in affinity to both metabolites. This is somewhat contradictory with the fact that PI binding can apparently be rescued by abrogating heme binding, indicating that PI binding has been kept throughout evolution. The authors mention a "barcode" of aminoacids necessary for PI binding, which all have been kept in Sfh5. Moreover, mutating one amino acid of this barcode allows a heme-binding-deficient mutant to regain heme binding to some extent, suggesting that it is PI binding that prevents heme binding in the heme-defective mutant. So what are the amino-acids of this barcode? Are they conserved, and if so, why?If the "resuscitable" PI-binding ability of Sfh5 is conserved, then it probably means that PI-binding is an important property of the protein.Also, the concentrations of heme and PI might be very different wherever Sfh5 is located, which might compensate for the different inherent affinities. Finally, the authors rightly acknowledge that "the formal possibility that Sfh5 donates heme to a highly privileged, and as-yet-unidentified, acceptor cannot be excluded", they do not acknowledge the possibility that Sfh5 binding to a third party factor (not necessarily a heme-acceptor) might decrease its affinity for heme in vivo. These points suggest that it could be premature to dismiss the possibility that heme/PI exchange might actually happen in vivo. This possibility does not alter the interest of the present study and can be addressed in further studies.

The barcodes they are referring to here is the PtdIns binding residues that are conserved in Sec14 and in all Sec14-like proteins -- including in all Sfh5 homologs. These are Y_68_ in Sfh5/R_65_ in Sec14, E_204_/E_207_, K_236_/K_239_, T_233_/T_236_, etc. These conserved residues are identified with a ▲ in Figure3—figure supplement 1.

Regarding the Y68 mutant that compromises the PtdIns-binding barcode, we really do not know how to interpret the fact that this specific mutation *restores* a small amount of heme binding to the Y175F mutant which is normally heme-less as it has lost the coordinating tyrosine. But, we felt we had to report the result because it was so unexpected. Perhaps H173 now coordinates the heme in this mutant (albeit very poorly). How and why this happens we have no good idea at the moment. We do not think there is a competition between PtdIns and heme binding, however. We hypothesized this to be the case initially but all data were most simply interpreted to the contrary. Evidence to that effect is covered in more detail in the response to reviewer 2 point 3.

- This manuscript talks to scientists with very different sets of expertise. The thorough characterization of the biophysical properties of Sfh5 could in general do with more complete explanations of the various methods in use. For instance:Soret spectra are calibrated against mock-expressing bacterial lysate. This information could go in the text or figure legend rather than the Material and methods section.

Revised as requested.

"The isostructural A10/T4 gating helices of Sfh1 and Sfh5 were positioned similarly across the opening to the internal cavity". The corresponding figure could benefit from better description (no labels on the helices).

The figure is now clarified by labeling the iso-structural gating helices.

"tyrosine coordination was consistent with the results of safranin T dye-reduction assays which estimated an unusually low thermodynamic reduction potential for Sfh5 heme (-330 ± 3 mV vs. NHE)." This could require better explanations (e.g. NHE is not described anywhere).

We now define NHE (normal hydrogen electrode).

Reviewer #2:[…]Essential revisions:1) The paper is technically sound but may be of limited impact due to the lack of data establishing a physiological function for heme binding. In this sense, significant efforts must be made to establish a role for heme binding. As it currently stands, the work may be more appropriate for a specialized audience of biochemists.

We concede the basic fact that, try as we might, we have yet to establish a physiological function for Sfh5 but the fact that it defines a novel clade of fungal hemoproteins is consistent with it having an important, although perhaps a boutique, function.

2) The authors should comment on how the PITP activity of the heme-binding mutants of Sfh5 compare to other Sec14-like PITPs? These mutants may have more PITP activity than WT Sfh5, but is the activity comparable to other more typical Sec14-type PITPs?

The specific activity of the heme-binding Sfh5 mutants is comparable to those of Sec14 and other yeast Sfh PITPs (Sfh1 is not an active PITP):

Sfh5 / Sec14: 0.1

Sfh5^Y175F^ / Sec14: 0.7

Sfh5^Y175F^ / Sfh2: 0.7

Sfh5^Y175F^ / Sfh3: 1.0

Sfh5^Y175F^ / Sfh4: 0.8

The only one that is relevant to this is the Sec14 comparison and that value is now stated in the text.

The following suggestions are outside the scope of the current study as presented. I hope the comments may prove useful to the authors as they continue their studies on this interesting protein.3) Do the authors think heme plays a regulatory role in Sfh5 phosphatidylinositol transfer activity? For instance, it is mentioned that only 1/3 of the protein is heme loaded. Does increasing heme synthesis decrease PITP activity in vivo? If the authors fed WT cells porphobilinogen (PBG), which would increase heme levels, or titrate aminolevulinic acid (ALA) in hem1∆ cells, is there a dose dependent decrease in Sfh5-related PITP activity in vivo? An experiment such as this demonstrating in vivo regulation of PITP activity by heme would greatly elevate the significance/impact of the current work.Note, in Saccharomyces cerevisiae, ALA synthase is not the rate-limiting enzyme for heme synthesis, as it is in other eukaryotes. Thus, to drive heme synthesis in WT cells, one must use PBG.

Why is only 30% of the protein occupied by heme? We suspect it is because *E. coli* does not make sufficient heme to occupy the amount of Sfh5 produced. We did test to see if supplementing the induction culture with α-ALA increased heme occupancy. It did not. We expound on this point in more detail in our response to reviewer 3 point 1 below.

Reviewer 2 suggests a clever and informative experiment and we are grateful for the input. We performed the version of this experiment outlined by the Editor in the Essential revisions for this paper. The experiment performed assessed whether Sfh5 over-expression in a heme-less *hem1∆* yeast strain cultured on ergosterol-containing medium could recapitulate phenotypic rescue of the *sec14ts* mutation as is seen for the heme-binding mutants, and whether this rescue could be abrogated by heme synthesis induced by ALA supplementation. We show these results in a revised Figure 7 – specifically a new Figure 7C. Sfh5 over-expression failed to rescue *sec14ts* defects at 37°C in heme-less cells. These data indicate WT Sfh5 that is not heme-bound is nonetheless an inactive PITP whereas a mutant defective in heme-binding has PITP activity. We interpret the data to indicate the heme coordinating residues in WT Sfh5 are incompatible with PtdIns binding while substitutions that result in heme-binding deficiency have the dual effects of compromising heme coordination and being compatible with PtdIns binding.

4) The authors make the intriguing comment that there may be a privileged acceptor that Sfh5 may donate heme to. Related to this point, I am wondering if heme reduction could induce its dissociation? I understand that the myoglobin transfer experiments were done with ferric heme. Could heme dissociate and transfer to myoglobin if it were reduced with dithionite?

We have performed this experiment using a more cumbersome Soret-based assay and find that dithionite reduction does not render Sfh-bound heme exchangeable in an apo-myoglobin loading assay. This result is now indicated in the text.

Sfh1 in some sense reminds me of yeast Dap1, a homolog of mammalian PGRMC1 and PGRMC2. Dap1 is a low potential hemoprotein (~-300 mV) that binds heme with a tyrosine ligand. It was thought that Dap1 might be a heme chaperone for ER localized Cytochrome P450s, although this is controversial. dap1∆ exhibits fluconazole and MMS sensitivity, in part due to defects in ergosterol biosynthesis, which requires a P450 enzyme. The authors may want to consider looking at fluconazole sensitivity, or defects in the ergosterol biosynthetic pathway. There was recently a paper in Nature describing role of PGRMCs in nuclear heme trafficking in adipocytes.

Thank you for the suggestion. The *sfh5∆* allele does not alter fluconazole sensitivity of yeast nor does it show any obvious genetic interactions with defects in the ergosterol biosynthetic pathway. But this general idea is one that merits attention.

Reviewer #3:[…]1) The affinity of Sfh5 for heme should be examined in more detail. This study suggests bound heme is not a readily exchangeable, but this is difficult to square with the statement in the discussion that only 1/3 of Sfh5s are heme-bound. The fraction of endogenous Sfh5 bound to heme should be determined and shown. It is important that these measurements be made with protein lacking a His-tag since this tag bind heme. If it is correct that bound heme is not readily exchangeable but only a fraction of the protein is heme-bound, some effort should be made to explain this. Is the on rate very slow?

We demonstrate that position of the His-tag in Sfh5 does not affect its heme-binding properties, and that a His-tagless version of Sfh5 retains:

1) heme-binding activity, and

2) the crystal structure unambiguously places the heme-binding pose in the internal hydrophobic pocket of Sfh5 far away from the His-tag.

Moreover, we find no evidence from extensive studies with other related His-tagged Sec14-like PITPs that the His-tag endows some artifactual heme-binding activity.

Why is only 30% of the protein occupied by heme? As addressed above for reviewer 2 point 3, we suspect it is because *E. coli* does not make sufficient heme to occupy the amount of Sfh5 produced. We did test to see if supplementing the induction culture with α-ALA increased heme occupancy. It did not for Sfh5 so it remains formally possible that the on-rate is slow. However, those results are also consistent with a model where heme is loaded co-translationally with a fast on-rate. It remains a formal possibility that, without sufficient available heme, a fraction of the protein subtly malfolds and cannot be loaded with heme once it goes off-pathway. We have tried several methods to strip purified Sfh5 of bound heme but the end result is always aggregation. We hope all will agree that the important questions relating to the mechanism of heme-loading, the kinetics of heme loading, etc are outside the scope of this work and could not be completed in a reasonable time frame.

2) The study demonstrates that heme bound by Sfh5 is accessible to small ligands like CN-, indicating that the heme is redox active, but does not explain how changes in the redox state of bound heme could be physiologically relevant. How could these changes be sensed by other proteins? Or how do the authors think bound heme could significantly contribute to cytoplasmic redox chemistry?

The questions posed by the reviewer are important and rather broad ones. However, without a physiological context from which to investigate these questions, we do not know how we can address the questions posed. We can only speculate. Given the HIP-HOP data our current hypothesis is that Sfh5 could function to detoxify an organic oxidant(s). Hemoproteins that exhibit such activities typically show very high specificity towards, and high specific activity against, their privileged substrates. But these proteins exhibit very low activities towards other substrates. So, we would need to go fishing for substrate – a daunting task with no clear prospects for success. We hope all will agree that the questions posed, while of critical importance, are simply not experimentally accessible at this time.

Reviewer #4:[…]Essential revisions:1) Introduction: The authors provide a lot of information about the lipid-dependent activities of the PITPs, given that their protein of interest (Sfh5) does not bind lipid and does not appear to function in the manner consistent with PITP function. Since it appears that the function of Sfh5 is unknown (which is interesting), perhaps this should be stated up front along with any information known about its biology in yeast and other eukaryotic homologs-then more briefly mention the functions of related Sec14-like PITPs and lipid-dependent activities as a possible function (i.e., present a hypothesis). The first paragraph of the Results section could be moved up to the Introduction.

Revised as requested.

2) Most of the work to show that Sfh5 binds heme was performed using protein expressed in bacteria, which produced heme-bound Sfh5 that copurifed during purification steps. One experiment is shown to link heme binding to Sfh5 in yeast cells (Figure 1C, lower right panel) but the data signal-to-noise is very low for the yeast protein compared to the bacterial protein and no control is provided to show the heme interaction in the yeast pull-down experiment is specific. Can growth conditions for bacterial or yeast grown be discovered that allow purification of heme-free Sfh5-or biochemical stripping/unfolding/repurification of heme-free protein-which can then be used in heme binding analysis via absorbance or ITC measurements and phosphatidylinositol binding assays?

Sfh5 is expressed at very low levels in yeast cells and purifying it from yeast in any quantity is a challenge. The question was whether Sfh5 purified from yeast came bound to heme and it does. We trust all will agree the weight of the data identify Sfh5 as a bona fide hemoprotein. We have tried hard to produce heme-free Sfh5 by various methods exactly for the purposes suggested by the reviewer. Those efforts have met with no success. The protein from which heme is extracted simply aggregates in all cases and, as indicated in our response to reviewer 3 point 1, the heme-free fraction of the recombinant Sfh5 is also inactive by any measure we have at our disposal. We also now add the experiment that shows heme-free WT Sfh5 produced in yeast devoid of heme is NOT resuscitated for PITP activity – unlike the heme-binding mutants (see response to reviewer 2, point 3). Our interpretation is that the mutations that abrogate heme binding have the added effect of permitting PtdIns-binding whereas the wild-type Tyr/His coordination motif is intrinsically nonpermissive for PtdIns-binding. Those new data are consistent with our initial interpretations.

3) There is no functional evidence presented to prove that Sfh5, which is expressed at low levels and shows no deletion functional phenotype, or its unique heme-binding properties are important in yeast or any other organism. Is it possible Sfh5 could be vestigial in yeast? Do related proteins from other organisms (i.e., those that evolutionarily precede yeast) express Sfh5 with higher expression and more robust heme-dependent functional activities. If Figure 8—figure supplement 2 provides the only insight into the potential functional relevance of Sfh5, the authors should consider moving it to a main figure.

Whether this is a vestigial gene or not is a teleological question we cannot confidently answer. We agree that assessing the functions of Sfh5 orthologs in other systems is a likely next step. But we hope all agree this is well beyond the scope of this work. However, the fact that it is conserved across significant evolutionary distances suggests to us this is not vestigial.

4) Crystallography: The authors show that Sfh5 is a homodimer using AUC; however, no dimerization was observed in the crystal structure. An issue related to this that should be addressed: when the sfh5_map_coeffs.mtz file provided by the authors is analyzed in CCP4, the Matthews_coef analysis (estimate of the number of asymmetric units in the unit cell) indicates there should be 4 molecules (e.g., two dimers?) in one unit cell with a solvent content ~52.2%. However, the authors solved the structure with 3 molecules in one unit cell. Other issues that should be addressed pertaining to the structure refinement include (1) the Ramachandran outliers in the validation report is 0.5%, which is inconsistent with 0% reported by the authors in the Table 1; (2) the percent of residues with favored Ramachandran is 86%, which can likely be improved; and (3) the R-factor gap (R_work_/R_free_) is large. Finally, the number of atoms in the protein (129) and ligand (5) reported by the authors in Table 1 needs to be corrected.

The AUC experiments were performed at physiological pH whereas the crystallization conditions were at pH 4.2 so the two results cannot be confidently compared. Regarding the Matthews_coef analysis, the calculated asymmetric unit content with 3 molecules gives a solvent content of 66.4% and with 4 molecules you get a solvent content of 55.1 %. Both are in the acceptable range for proteins. The SAD electron density showed only 3 polypeptide chains per ASU with no additional chain density or gaps in the packing identified.

1) Regarding the inconsistency of the validation report vs. Table 1, we thank reviewer 4 for the careful examination. We mistakenly rounded to the whole and it is corrected now.

2) Regarding improvement of the favored Ramachandran, as described above, only the density for chain B was well-ordered. The resultant model for chain B has 100% fit to electron density based on RSRZ values, and no Ramachandran outliers. Chain B model was used for all analysis and interpretation, as well as a guide to build chains A and C, which have large regions of poor density. We now explicitly state this in the text. These regions are the main contributors to the Ramachandran outliers. Imposing stricter geometry restrains by changing weights in the refinement leads to greater gap between R_free_ and R_work_.

3) Regarding the R-factor gap, as stated above, this was a challenging structure to solve and build as chains A and C show large regions of disordered density in the original SAD maps and with model phases. Multiple attempts were made to rebuild the model and multiple data sets were sampled – all sharing the same problems. We chose the best R_work_ and R_free_ combination that we could obtain and stopped the refinement once R_free_ started to increase when we attempted to fix geometry further (as suggested above).

4) The values have been corrected in the Table 1. Thank you for pointing out this error.

5) Figure 7: The authors prepared apo-myoglobin by methy-ethyl ketone extraction of heme from equine skeletal myoglobin for use in heme-transfer assays to determine if Sfh5 can exchange heme via detection of heme transfer to apo-myoglobin. However, they do not show a critical positive control showing that free heme can bind to their prepared apo-myoglobin to demonstrate the protein is functionally capable of binding heme.

That control is now included in a new Figure 8.